# Efficient Dynamics Modeling in Interactive Environments with Koopman Theory

**Arnab Kumar Mondal** *
Mila, McGill University

**Siba Smarak Panigrahi**
Mila, McGill University

**Sai Rajeswar**
ServiceNow Research

**Kaleem Siddiqi**
Mila, McGill University

**Siamak Ravanbakhsh**
Mila, McGill University

## Abstract

The accurate modeling of dynamics in interactive environments is critical for successful long-range prediction. Such a capability could advance Reinforcement Learning (RL) and Planning algorithms, but achieving it is challenging. Inaccuracies in model estimates can compound, resulting in increased errors over long horizons. We approach this problem from the lens of Koopman theory, where the nonlinear dynamics of the environment can be linearized in a high-dimensional latent space. This allows us to efficiently parallelize the sequential problem of long-range prediction using convolution while accounting for the agent's action at every time step. Our approach also enables stability analysis and better control over gradients through time. Taken together, these advantages result in significant improvement over the existing approaches, both in the efficiency and the accuracy of modeling dynamics over extended horizons. We also show that this model can be easily incorporated into dynamics modeling for model-based planning and model-free RL and report promising experimental results.

## 1 Introduction

The ability to predict the outcome of an agent's action over long horizons is a crucial unresolved challenge in Reinforcement Learning (RL) (Sutton & Barto, 2018; Mnih et al., 2015; Silver et al., 2017; Mnih et al., 2016). This is especially important in model-based RL and planning, where deriving a policy from the learned dynamics models allows one to efficiently accomplish a wide variety of tasks in an environment (Du & Narasimhan, 2019; Hafner et al., 2020; Sikchi et al., 2021; Hansen et al., 2022; Schrittwieser et al., 2020; Jain et al., 2022). In fact, state-of-the-art model-free techniques also rely on dynamics models to learn a better representation for downstream value prediction tasks (Schwarzer et al., 2020). Thus, obtaining accurate long-range dynamics models in the presence of input and control is crucial. In this work, we leverage techniques and perspectives from *Koopman theory* (Koopman, 1931; Mauroy et al., 2020; Koopman & Neumann, 1932; Brunton et al., 2021) to address this key problem in long-range dynamics modeling of interactive environments. The application of Koopman theory allows us to linearise a nonlinear dynamical system by creating a bijective mapping to linear dynamics in a possibly infinite dimensional space of *observables*.[1] Conveniently, a deep neural network can learn to produce such a mapping, enabling a reliable approximation of the non-linear dynamics in the finite dimension of a linear latent space so that only a finite subset of the most relevant (complex-valued) Koopman observables need to be tracked.

We show that this linearization has two major benefits: 1) The eigenvalues of the linear latent-space operator are directly related to the stability and expressive power of the controlled dynamics model. Furthermore, spectral Koopman theory provides a unifying perspective over several existing directions on stabilizing gradients through time, including long-range sequence modeling using state-space models (Gu et al., 2020; 2022b;a; Gupta et al., 2022), and the use of unitary matrices (Arjovsky et al., 2016) in latent dynamic models; 2) This can help to avoid a computational bottleneck due

---

*Correspondence: arnab.mondal@mila.quebec. Work done while at ServiceNow Research

[1]The term observables can be misleading as it refers to the latent space in the machine learning jargon.

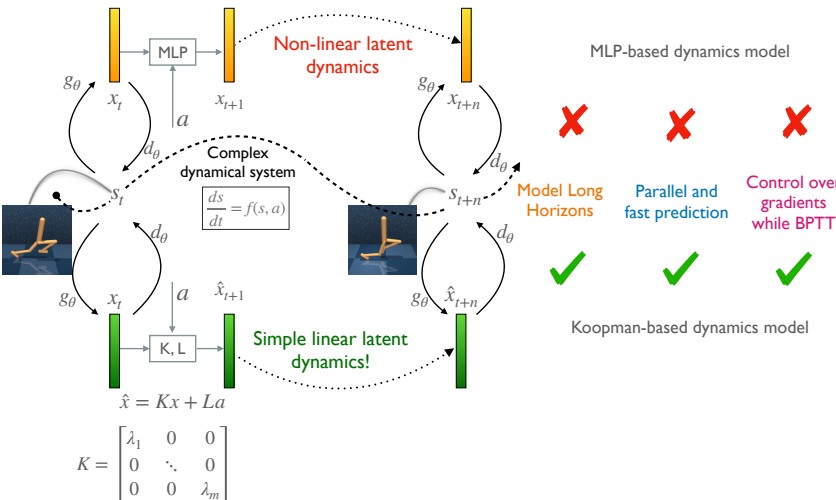

Figure 1: A comparison of our Koopman-based linear dynamics model with a non-linear MLP-based dynamics model. The Diagonal Koopman formulation allows for modeling longer horizons efficiently with control over gradients. Here BPTT stands for Backpropagation Through Time.

to the sequential nature of dynamics. This is achieved by a reformulation using convolution and diagonalization of the linear operator. In other words, one can perform efficient parallel training of the model over time steps, despite dealing with a controlled dynamical system.

Our experimental results in offline-RL datasets demonstrate the effectiveness of our approach for reward and state prediction over a long horizon. In particular, we report competitive results against dynamics modeling baselines that use Multi-Layer Perceptrons (MLPs), Gated Recurrent Units (GRUs), Transformers and Diagonal State Space Models Gu et al. (2022a) while being significantly faster. Finally, we also present encouraging results for model-based planning and model-free RL with our Koopman-based dynamics model.

In summary, our contributions are:

1. We formulate an efficient linear latent dynamics model from *Koopman theory* that can be parallelized during training for long-range dynamics modeling in interactive environments.
2. We demonstrate the use of the spectral decomposition of the Koopman matrix, which enables better control over gradients through time and accelerated training.
3. We show that our technique serves as a preferred alternative for existing dynamics models in planning and RL algorithms, offering enhanced performance, speed and training stability.

## 2 PRELIMINARIES

### 2.1 KOOPMAN THEORY FOR DYNAMICAL SYSTEMS

In the context of non-linear dynamical systems, the *Koopman operator*, a.k.a. the Koopman-von Neumann operator, is a linear operator for studying the system's dynamics. The Koopman operator is defined as a linear transformation that acts on the space of functions or observables of the system, known as the observables space $\mathcal{F}$. For a non-linear discrete or continuous time dynamical system

$$x_{t+1} = F(x_t) \quad \text{or} \quad \frac{dx}{dt} = f(x)$$

the Koopman operator $\mathcal{K} : \mathcal{F} \to \mathcal{F}$, is defined as $\mathcal{K}g \cong g \circ F$ where $\mathcal{F}$ is the set of all functions or observables that form an infinite-dimensional Hilbert space. In other words, for every function $g : X \to \mathbb{R}$ belonging to $\mathcal{F}$, where $x_t \in X \subset \mathbb{R}^n$, we have

$$(\mathcal{K}g)(x_t) = g(F(x_t)) = g(x_{t+1}).$$

The infinite dimensionality of the Koopman operator presents practical limitations. Addressing this, we seek to identify a subspace $\mathcal{G} \subset \mathcal{F}$, spanned by a finite set of observables $g_1, \ldots, g_m$ where typically $m \gg n$, that approximates invariance under the Koopman operator.

Constraining the Koopman operator on this invariant subspace results in a finite-dimensional linear operator $K \in \mathbb{C}^{m \times m}$, called the *Koopman matrix*, which satisfies $g(x_{t+1}) = Kg(x_t)$. Usually, the base observation functions $\{g_i\}_i$ are hand-crafted using knowledge of the system's underlying physics. However, data-driven methods have recently been proposed to learn the Koopman operator by representing the base observations or their eigenfunction basis using deep neural networks, where a decoder reconstructs the input from linear latent dynamics (*e.g.*, Lusch et al., 2018; Champion et al., 2019). Linearizing the dynamics allows for closed-form solutions for the predictions of the dynamical system, as well as stability analysis based on the eigendecomposition of the Koopman matrix Fan et al. (2022); Yi & Manchester (2023).

The observation functions $\{g_i\}_i$ spanning an invariant subspace can always be chosen to be *eigenfunctions* $\{\phi_i\}_i$ of the Koopman operator, *i.e.*, $\mathcal{K}\phi_i(x) = \lambda_i\phi_i(x), \forall x$. With this choice of observation functions, the Koopman matrix becomes diagonal,

$$K_D = \operatorname{diag}(\lambda_1, \ldots, \lambda_m), \quad \lambda_i \in \mathbb{C} \quad \forall i. \tag{1}$$

By using a diagonal Koopman matrix, we are effectively offloading the task of learning the suitable eigenfunctions of the Koopman operator that form an invariant subspace to the neural network encoder. Additionally, for a continuous time input, to be able to model higher-order frequencies in the eigenspectrum and provide better approximations, one could adaptively choose the most relevant eigenvalues and eigenfunctions for the current state (Lusch et al., 2018). This means $g(x_{t+1}) = K_D(\lambda(g(x_t)))g(x_t)$, where $\lambda : \mathcal{G} \to \mathbb{C}^m$ can be a neural network.

## 2.2 APPROXIMATE KOOPMAN WITH CONTROL INPUT

The Koopman operator can be extended to non-linear *control* systems, where the state of the system is influenced by an external control input $u_t$ such that $x_{t+1} = F(x_t, u_t)$ or $\frac{dx}{dt} = f(x, u)$. Simply treating the pair $(x, u)$ as the input $x$, the Koopman operator becomes $(\mathcal{K}g)(x_t, u_t) = g(F(x_t, u_t), u_{t+1}) = g(x_{t+1}, u_{t+1})$. If the effect of the control input on the system's dynamics is linear, i.e., the *control affine* setting, we have

$$f(x, u) = f_0(x) + \sum_{i=1}^{m} f_i(x)u_i.$$

assuming a finite number of eigenfunctions. For such control-affine system, one could show that the Koopman operator is *bilinearized* (Brunton et al., 2021; Bruder et al., 2021):

$$g(x_{t+1}) = \mathcal{K}(u)g(x_t) = (\mathcal{K}_0 + \sum_{i=1}^{m} u_i\mathcal{K}_i)g(x_t).$$

More generally, one could make the Koopman operator a function of $u_t$, so that the resulting Koopman matrix satisfies $g(x_{t+1}) = K(u_t)g(x_t)$. This is the approach taken by Weissenbacher et al. (2022) to model symmetries of dynamics in offline RL. When combined with the diagonal Koopman matrix of Eq. (1), and fixed modulus $|\lambda_i| = 1 \,\forall i$, the Koopman matrix becomes a product of $SO(2)$ representations, and the resulting latent dynamics model, resembles a special setting of the symmetry-based approach of Mondal et al. (2022), where group representations are used to encode states and state-dependent actions.

An alternate approach to approximating the Koopman operator with a control signal assumes a decoupling of state and control observables $g(x, u) = [g(x), f(u)] = [g_1(x), \ldots, g_m(x), f_1(u), \ldots, f_n(u)]$ Brunton et al. (2021); Bruder et al. (2019). This gives rise to a simple linear evolution:

$$g(x_{t+1}) = Kg(x_t) + Lf(u_t) \tag{2}$$

where $K \in \mathbb{C}^{m \times m}$ and $L \in \mathbb{C}^{m \times l}$ are matrices representing the linear dynamics of the state observables and control input, respectively. We build on this approach in our proposed methodology. Although, as shown in Brunton et al. (2016; 2021); Bruder et al. (2019), even assuming a linear control term $Lu_t$ can perform well in practice, we use neural networks for both $f$ and $g$. Our

rationale for choosing this formulation is that the additive form of Eq. (2) combined with the *diagonalized Koopman matrix* of Eq. (1), enable fast parallel training across time-steps using a convolution operation where the convolution kernel can be computed efficiently. Moreover, this setting is amenable to the analysis of gradient behaviour, as discussed in Section 3.2.1. For a more comprehensive overview of Koopman theory in the context of controlled and uncontrolled dynamics, we direct readers to Brunton et al. (2021); Mauroy et al. (2020).

# 3 PROPOSED MODEL

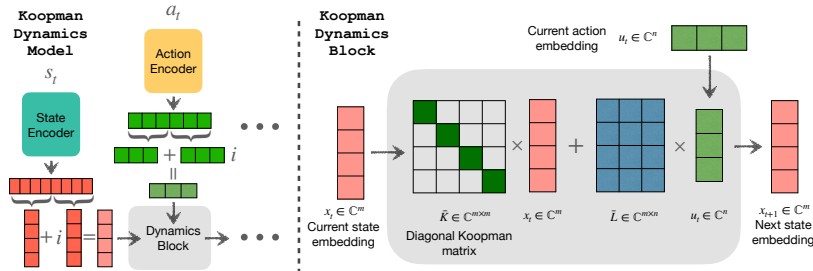

Figure 2: A schematic of the latent Koopman dynamics model. Both actions and initial state embedding are encoded into a latent space in complex ($\mathbb{C}$) domain before passing through the Koopman dynamics block. (see Appendix J for an efficient Jax implementation of the model)

## 3.1 LINEAR LATENT DYNAMICS MODEL

Our task in dynamics modeling is to predict the sequence of future states $s_{t+1}, ...., s_{t+\tau}$ given a starting state $s_t$ and some action sequence $a_t, a_{t+1}, ..., a_{t+\tau-1}$, assuming a Markovian system. Following the second approach described in Section 2.2, we assume that state and control observables are decoupled, and use neural network encoders[2] to encode both states and actions

$$x_t = g_\theta(s_t) \quad \text{and} \quad u_t = f_\phi(a_t). \tag{3}$$

Due to this decoupling, the continuous counterpart of latent space dynamics Eq. (2) is

$$\frac{\mathrm{d}x}{\mathrm{d}t} = Kx(t) + Lu(t), \tag{4}$$

and the solution is given by $x(s) = e^{Ks}x(0) + \int_0^s e^{K(s-t)}Lu(t)\mathrm{d}t$ where $e^{Ks}$ is a matrix exponential. One could discretize this to get $\hat{x}_{t+1} = \bar{K}x_t + \bar{L}u_t$, where $\bar{K}$ and $\bar{L}$ are obtained by *Zero-Order Hold (ZOH)* discretization of the continuous-time equation (Iserles, 2009):

$$\bar{K} = \exp(\Delta t K) \qquad \bar{L} = (K)^{-1}(\exp(\Delta t K) - 1)L$$

In practice, we assume that observations are sampled uniformly in time and use a learnable time step parameter $\Delta t$. Unrolling the discrete version of the dynamics for $\tau$ steps, we get:

$$[\hat{x}_{t+1}, \cdots, \hat{x}_{t+\tau}] = [\bar{K}, \cdots, \bar{K}^\tau]x_t + [c_t, \cdots, c_{t+\tau-1}] \underbrace{\begin{bmatrix} I & \bar{K} & \cdots & \bar{K}^{\tau-1} \\ 0 & I & \cdots & \bar{K}^{\tau-2} \\ \vdots & \vdots & \ddots & \vdots \\ 0 & 0 & \cdots & I \end{bmatrix}}_{\Gamma}, \tag{5}$$

where $c_{t+k} = \bar{L}u_{t+k}$, $\Gamma$ encodes the effect of each input on all future observables, and the ˆ in $\hat{x}$ distinguishes the prediction from the ground truth observable $x$. (see Appendix B)

We can then recover the predicted state sequence by inverting the function $g_\theta$, using a decoder network $\hat{s}_{t+k} = d_\xi(\hat{x}_{t+k})$. Loss functions in both input and latent space can be used to learn the encoder,

---

[2]We use a CNN for pixel input and an MLP for state-based input and actions.

decoder, and latent dynamics model. We refer to these loss functions as the *Koopman consistency loss* and the *state-prediction loss*, respectively:

$$\mathcal{L}_{\text{consistency}} = \sum_{k=1}^{\tau} \|\hat{x}_{t+k} - x_{t+k}\|_2^2 \qquad \mathcal{L}_{\text{state-pred}} = \sum_{k=1}^{\tau} \|d_\xi(\hat{x}_{t+k}) - s_{t+k}\|_2^2 \qquad (6)$$

## 3.2 Diagonalization, Efficiency and Stability of the Koopman Operator

Since the set of diagonalizable matrices is dense in $\mathbb{C}^{m \times m}$ and has a full measure, we can always diagonalize the Koopman matrix as shown in Eq. (1). Next, we show that this diagonalization can be leveraged for efficient and parallel forward evolution and training.

To unroll the latent space using Eq. (5) we need to perform matrix exponentiation and dense multiplication. Using a diagonal Koopman matrix $\bar{K} = \text{diag}(\bar{\lambda}_1, \ldots, \bar{\lambda}_m)$, this calculation is reduced to computing an $m \times (\tau + 1)$ complex-valued *Vandermonde matrix* $\Lambda$, where $\Lambda_{i,j} = \bar{\lambda}_i^j$, and doing a row-wise circular convolution of this matrix with a sequence of vectors.

Recall that $x, c \in \mathbb{C}^m$ are complex vectors. We use superscript to index their elements – *i.e.*, $x_t = [x_t^1, \ldots, x_t^m]$. With this notation, the $i$-th element of the predicted Koopman observables for future time steps is given by (see Appendix B for the derivation):

$$[\hat{x}_{t+1}^i, \ldots, \hat{x}_{t+\tau}^i] = [\bar{\lambda}_i, \ldots, \bar{\lambda}_i^\tau] x_t^i + [1, \bar{\lambda}_i, \ldots, \bar{\lambda}_i^{\tau-1}] \circledast [c_t^i, \ldots, c_{t+\tau-1}^i] \qquad (7)$$

where $\circledast$ is circular convolution with zero padding. The convolution can be efficiently computed for longer time steps using Fast Fourier Transform. (Brigham, 1988) (see Appendix J)

### 3.2.1 Gradients Through Time

We show how we can control the behavior of gradients through time by constraining the real part of the eigenvalues of the Diagonal Koopman matrix. Let $\mu$ and $\omega$ refer to the real and imaginary part of the Koopman eigenvalues – that is $\lambda_j = \mu_j + i\omega_j$.

**Theorem 3.1.** *For every time step $k \in \{1, .., \tau\}$ in the discrete dynamics, the norm of the gradient of any loss at $k$-step given by $\mathcal{L}_k$ with respect to latent representation at time step $t$ given by $x_t$ is a scaled version of the norm of the gradient of the same loss by $x_{t+k}$, where the scaling factor depends on the exponential of the real part of the Koopman eigenvalues, that is:*

$$|\frac{\partial \mathcal{L}_k}{\partial x_t^j}| = e^{k\Delta t \mu_j} |\frac{\partial \mathcal{L}_k}{\partial x_{t+k}^j}| \quad \forall j \in \{1, .., m\}.$$

*and similarly, for all $l \leq k$, the norm of the gradient of $\mathcal{L}_k$ with respect to the control input at time step $t + l - 1$ given by is $c_{t+l-1}^j$ is a scaled version of the norm of the gradient of $\mathcal{L}_k$ by $x_{t+k}$, where the scaling factor depends on the exponential of the real part of the Koopman eigenvalues, that is:*

$$|\frac{\partial \mathcal{L}_k}{\partial c_{t+l-1}^j}| = e^{(k-l)\Delta t \mu_j} |\frac{\partial \mathcal{L}_k}{\partial \hat{x}_{t+k}^j}| \quad \forall j \in \{1, .., m\}$$

A proof of this theorem is provided in Appendix A. The theorem implies that the amplitude of the gradients from a future time step scales exponentially with the real part of each diagonal Koopman eigenvalue. We use this theorem for better initialization of the diagonal Koopman matrix as explained in the next section.

### 3.2.2 Initialization of the Eigenspectrum

The *imaginary* part of the eigenvalues of the diagonal Koopman matrix captures different frequency modes of the dynamics. Therefore, it is helpful to initialize them using increasing order of frequency, that is, $\omega_j := \alpha j \pi$, for some constant $\alpha$, to cover a wide frequency range.

From Theorem A.1, we know that the real part of the eigenvalues impacts the gradient's behavior through time. To avoid vanishing gradients and account for prediction errors over longer horizons, one could eliminate the real part $\mu_j := 0$. This choice turns the latent transformations into blocks of 2D rotations (Mondal et al., 2022). This is also related to using Unitary Matrices to avoid

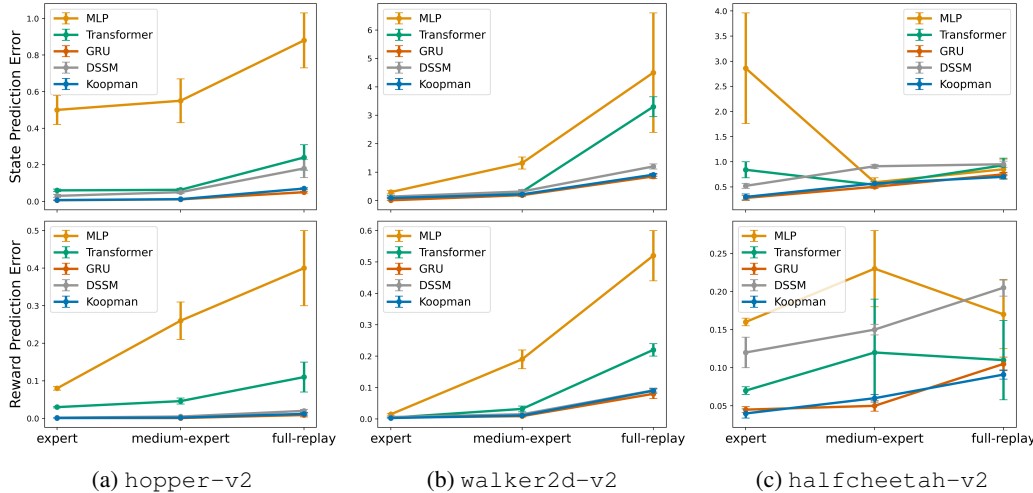

(a) `hopper-v2`      (b) `walker2d-v2`      (c) `halfcheetah-v2`

Figure 3: Forward state and reward prediction error in Offline Reinforcement Learning environments. We consider five dynamics modeling techniques and perform this prediction task over a **horizon of 100 environment steps**. The results are over 3 runs. Our Koopman-based method is competitive with the best performing GRU baseline while being $2\times$ faster. See Appendix E for exact numerical values.

vanishing or exploding gradients in Recurrent Neural Networks (Arjovsky et al., 2016). However, intuitively, we might prefer to prioritize closer time steps. This can be done using small negative values *e.g.*, $\mu_j \in \{-0.1, -0.2, -0.3\}$. An alternative to having a fixed real part is to turn it into a bounded learnable parameter $\mu_j \in [-0.3, -0.1]$. We empirically found $\mu_j := -0.2 \ \forall j$, and $\omega_j := j\pi$ to be good choices and use this as our default initialization. In Appendix H we report the results of an ablation study with different initialization schemes.

## 4 EXPERIMENTS

### 4.1 LONG-RANGE FORWARD DYNAMICS MODELING WITH CONTROL

We model the non-linear controlled dynamics of MuJoCo environments (Todorov et al., 2012) using our Koopman operator. We choose a standard MLP-based latent non-linear dynamics model with two linear layers followed by a ReLU non-linearity as one of our baselines. To make it easier to compare with Koopman dynamics model, we use state embeddings of the same dimensionality using $g_\theta$, which is an MLP, as described in Section 3.1. This embedding is then fed to the MLP-based latent dynamics model or the diagonal Koopman operator. This approach is widely used in RL for dynamics modeling (Sikchi et al., 2021; Hansen et al., 2022; Schrittwieser et al., 2020; Schwarzer et al., 2020). To further compare our model with more expressive alternatives that can be trained in parallel, we design a causal Transformer (Vaswani et al., 2017; Chen et al., 2021) based dynamics model, which takes a masked sequence of state-actions

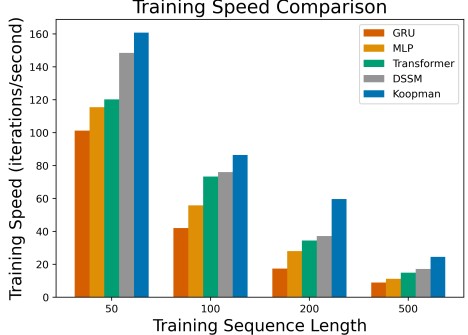

Figure 4: Training speed in iterations/second ($\uparrow$) for the state prediction task using different dynamics model on `halfcheetah-expert-v2`. Each iteration consists of one gradient update of the entire model using a mini-batch of 256 in A100 GPU. See Table 3 for exact numerical values.

and outputs representations that are used to predict next states and rewards.[3] Following the success of Hafner et al. (2020), we also design a GRU-based dynamics model that also takes a masked sequence

---

[3] All the states after the starting state are masked, but actions are fed to the model for all future time-steps.

of state-actions to output representations for state and reward prediction. Finally, we use the same strategy to also design a DSSM-based Gu et al. (2022a) dynamics model. To ensure a fair comparison in terms of accuracy and runtime, we maintain an equal number of trainable parameters for all the aforementioned models. (see Appendix F for more details)

For the forward dynamics modeling experiments, we use the D4RL Fu et al. (2020) dataset, which is a popular offline-RL environment. We select offline datasets of trajectories collected from three different popular Gym-MuJoCo control tasks: `hopper`, `walker`, and `halfcheetah`. These trajectories are obtained from three distinct quality levels of policies, offering a range of data representing various levels of expertise in the task: expert, medium-expert, and full replay. We divide the dataset of 1M samples into 80:20 splits for training and testing, respectively. To train the dynamics model, we randomly sample trajectories of length $\tau$ from the training data, where $\tau$ is the horizon specified during training. We test our learned dynamics model for a horizon length of 100 by randomly sampling 50,000 trajectories of length 100 from the test set. We use our learned dynamics model to predict the ground truth state sequence given the starting state and a sequence of actions.

**Training stability**    We can only train the MLP-based dynamics model for 10-time steps across all environments. Using longer sequences often results in exploding gradients and instability in training due to backpropagation through time (Sutskever, 2013). The alternative of using `tanh` in MLP-based models can lead to vanishing gradients. In contrast, our diagonal Koopman operator handles long trajectories without encountering vanishing or exploding gradients, in agreement with our discussion in Section 3.2.1.

### 4.1.1    STATE AND REWARD PREDICTION

In Fig. 3, we evaluate our model's accuracy in state and reward prediction over long horizons involving 100 environment steps. For this experiment, we add a reward predictor and jointly minimize the reward prediction loss and the state prediction loss. We set the weight of the consistency loss in the latent space to $0.001$. We see that for longer horizon prediction, our model is considerably better in predicting rewards and states accurately for `hopper`, `walker`, and `halfcheetah` in comparison to MLP, Transformer and DSSM based model while being competitive with GRU-based model. Furthermore, Fig. 4 empirically verifies that our proposed Koopman dynamics model is significantly faster than an MLP, GRU, DSSM and Transformer based dynamics model. The results also reveal the trend that our model's relative speed-up over baselines grows with the length of the horizon.

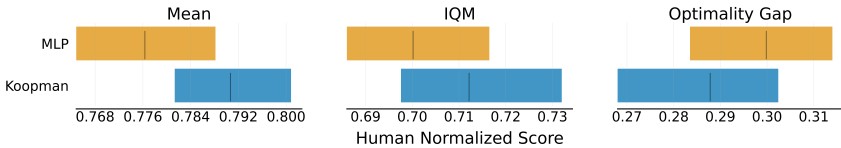

Figure 5: Comparison of our Koopman-based dynamics model (with a horizon of 20) and an MLP-based dynamics model of vanilla TD-MPC (Hansen et al., 2022). The results are over 5 random seeds for each environment. Higher Mean & IQM and lower Optimality Gap is better.

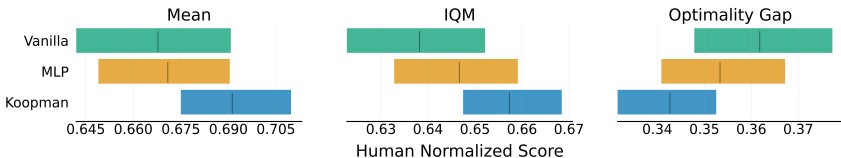

Figure 6: Comparison of vanilla SAC (Haarnoja et al., 2018) and its integration with an MLP-based (SPR) and a Koopman-based dynamics model for incorporating self-predictive representations in the DeepMind Control Suite. The results are over 5 random seeds for each environment. Higher Mean & IQM and lower Optimality Gap is better.

### 4.2 KOOPMAN DYNAMICS MODEL FOR RL AND PLANNING

We now present promising results from integrating the diagonal Koopman model into two areas: (I) model-based planning, and (II) model-free reinforcement learning (RL). In the latter, the dynamics model enhances representation learning. Both approaches aim to solve continuous control tasks in a data-efficient manner. We provide a brief summary of these methods and their modifications with the Koopman dynamics model in Appendix C. While the same dynamics model can also be used for (III) model-based RL, we leave that direction for future work.

**Evaluation Metric**   To assess the performance of our algorithm, we calculate the average episodic return at the conclusion of training and normalize it with the maximum score, i.e., 1000. However, due to significant variance over different runs, relying solely on mean values may not provide reliable metrics. To address this, Agarwal et al. (2021) suggests using bootstrapped confidence intervals (CI) with stratified sampling, which is particularly suitable for small sample sizes, such as the five runs per environment in our case. By utilizing bootstrapped CI, we obtain interval estimates indicating the range within which an algorithm's aggregate performance is believed to fall and report the Interquartile Mean (IQM). The IQM represents the mean across the middle $50\%$ of the runs, providing a robust measure of central tendency. Furthermore, we calculate the Optimality Gap (OG), which quantifies the extent to which the algorithm falls short of achieving a normalized score of 1.0. [4]

#### 4.2.1 MODEL-BASED PLANNING

We utilize TD-MPC (Hansen et al., 2022) as the baseline for our model-based planning approach. TD-MPC combines the benefit of a model-free approach in long horizons by learning a value function while using an MLP-based "Task-Oriented Latent Dynamics" (TOLD) model for planning over shorter horizons. To assess the effectiveness of our proposed dynamics model, we replace TOLD with our diagonal Koopman operator. Subsequently, we trained the TD-MPC agent from scratch using this modified Koopman TD-MPC approach. See Appendix C.4 for details on this adaptation. To evaluate the performance of our variation, we conducted experiments in four distinct environments: `Quadruped Run`, `Quadruped Walk`, `Cheetah Run`, and `Acrobot Swingup`. We run experiments in the state space and compare them against vanilla TD-MPC.

Our Koopman dynamics model was trained with a horizon of 20-time steps. We observe that TD-MPC suffers from training instability, performance degradation and high variance when using 20-time steps. Consequently, we opted to use a horizon of 5-time steps for the vanilla approach. Fig. 5 suggests that the Koopman TD-MPC outperforms the MLP-based baseline while being more stable.

#### 4.2.2 MODEL-FREE RL

Current data-efficient model-free RL approaches like (SPR; Schwarzer et al., 2020) use dynamics modeling as an auxiliary task to improve representation learning. We apply our dynamics model to this model-free RL framework, where we use an adaptation of SPR as our baseline. This adaptation uses an MLP-based dynamics model rather than the original CNN used in SPR in order to eliminate the advantage of the original SPR for image inputs. SPR uses an auxiliary consistency loss similar to Eq. (6), as well as augmentations and other techniques from Grill et al. (2020) to prevent representation collapse. It also improves the encoder's representation quality by making it invariant to the transformations that are not relevant to the dynamics using augmentation; see also Srinivas et al. (2020); Yarats et al. (2021). To avoid overconstraining the latent space using the dynamics model, SPR uses *projection heads*. Similarly, we encode the Koopman observables using a projection layer above the representations used for Q-learning. We provide more details on the integration of the Koopman dynamics model into the SPR framework in Appendix C.5.

We perform experiments on five different environments from the DeepMind Control (DMC) Suite (Tunyasuvunakool et al., 2020; Tassa et al., 2018) (`Ball in Cup Catch`, `Cartpole Swingup`, `Cheetah Run`, `Finger Spin`, and `Walker Walk`) in pixel space, and compare it with an MLP-based dynamics model (SPR) and no dynamics model, i.e., SAC (Haarnoja et al., 2018) as baselines. Figure 6 demonstrates that for a horizon of 5 time steps Koopman-based model outperforms the SAC plus MLP-based dynamics model baseline while being more efficient.

---

[4]A smaller Optimality Gap indicates superior performance.

## 5    RELATED WORK

**Diagonal state space models (DSSMs).**    DSSMs (Gu et al., 2022a; Gupta et al., 2022; Mehta et al., 2022) have been shown to be an efficient alternative to Transformers for long-range sequence modeling. DSSMs with certain initializations have been shown Gu et al. (2022a) to approximate convolution kernels derived for long-range memory using the HiPPO (Gu et al., 2020) theory. This explains why DSSMs can perform as well as structured state space models (S4) (Gu et al., 2022b), as pointed out by Gupta et al. (2022). Inspired by the success of gating in transformers (Hua et al., 2022), Mehta et al. (2022) introduced a gating mechanism in DSS to increase its efficiency further. However, a DSSM is a non-linear model used for sequence modeling and functions differently from our Koopman-based dynamics model, as explained in Appendix D.

**Probabilistic latent spaces.** Several early papers use approximate inference with linear and non-linear latent dynamics so as to provide probabilistic state representations (Haarnoja et al., 2016; Becker et al., 2019; Shaj et al., 2021; Hua et al., 2022; Fraccaro et al., 2017). A common theme in several of these articles is the use of Kalman filtering, using variational inference or MCMC, for estimating the posterior over the latents and mechanisms for disentangled latent representations. While these methods vary in their complexity, scalability, and representation power, similar to an LSTM or a GRU, they employ recurrence and, therefore, cannot be parallelized.

**Koopman Theory for Control.**    The past work in this area has solely focused on learning the Koopman operator or discovering representations for Koopman spectrum for control (Shi & Meng, 2022; Watter et al., 2015). Korda & Mezić (2018) extends the Koopman operator to controlled dynamical systems by computing a finite-dimensional approximation of the operator. Furthermore, they demonstrate the use of these predictors for model predictive control (MPC). Han et al. (2020) tasks a data-driven approach to use neural networks to represent the Koopman operator in controlled dynamical system setting. Kaiser et al. (2021) introduces a method using Koopman eigenfunctions for effective linear reformulation and control of nonlinear dynamical systems. Song et al. (2021) proposes deep Koopman reinforcement learning that learns control tasks with a small amount of data. However, none of these works focus on the training speed and stability for predicting longer horizon.

## 6    CONCLUSION

Dynamics modeling has a key role to play in sequential decision-making: at training time, having access to a model can lead to sample-efficient training, and at deployment, the model can be used for planning. Koopman theory is an attractive foundation for building such a dynamics model in Reinforcement Learning. The present article shows that models derived from this theoretical foundation can be made computationally efficient. We also demonstrate how this view gives insight into the stability of learning through gradient descent in these models. Empirically, we show the remarkable effectiveness of such an approach in long-range dynamics modeling and provide some preliminary yet promising results for model-based planning and model-free RL.

## 7    LIMITATIONS AND FUTURE WORK

Although our proposed Koopman-based dynamics model has produced promising results, our current treatment has certain limitations that motivate further investigation. Our current model is tailored for deterministic environments, overlooking stochastic dynamics modeling. We intend to expand our research in this direction, based on developments around *stochastic* Koopman theory (Mezić, 2005; Wanner & Mezic, 2022), where Koopman observables become random variables, enabling uncertainty estimation in dynamics. Secondly, although our state prediction task yielded impressive results, we have identified areas for further growth, particularly in its applications to RL and planning.

We hypothesize that the distribution shift during training and the objective change in training and evaluation hurt the performance of our Koopman dynamics model. We also plan to conduct a more comprehensive study involving a wider range of tasks and environments. Additionally, we aim to explore the compatibility of our model with different reinforcement learning algorithms to showcase its adaptability. Finally, we believe that the diagonal Koopman operator holds promise for model-based RL, where robust dynamics modeling over long horizons is precisely what is currently missing. We intend to investigate this direction, potentially unlocking new avenues for advancement.

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

## A  PROOF OF THEOREM 3.1

**Theorem A.1.** *For every time step $k \in \{1, .., \tau\}$ in the discrete dynamics, the norm of the gradient of any loss at $k$-step given by $\mathcal{L}_k$ with respect to latent representation at time step $t$ given by $x_t$ is a scaled version of the norm of the gradient of the same loss by $x_{t+k}$, where the scaling factor depends on the exponential of the real part of the Koopman eigenvalues, that is:*

$$|\frac{\partial \mathcal{L}_k}{\partial x_t^j}| = e^{k\Delta t \mu_j}|\frac{\partial \mathcal{L}_k}{\partial x_{t+k}^j}| \quad \forall j \in \{1, .., m\}.$$

*and similarly, for all $l \leq k$, the norm of the gradient of $\mathcal{L}_k$ with respect to the control input at time step $t + l - 1$ given by is $c_{t+l-1}^j$ is a scaled version of the norm of the gradient of $\mathcal{L}_k$ by $x_{t+k}$, where the scaling factor depends on the exponential of the real part of the Koopman eigenvalues, that is:*

$$|\frac{\partial \mathcal{L}_k}{\partial c_{t+l-1}^j}| = e^{(k-l)\Delta t \mu_j}|\frac{\partial \mathcal{L}_k}{\partial \hat{x}_{t+k}^j}| \quad \forall j \in \{1, .., m\}$$

*Proof.* We now provide a proof sketch of Theorem 3.1. As $\lambda_j = \mu_j + i\omega_j$, discretizing the diagonal matrix (using ZOH) and taking its $k$-th power gives us $\bar{K}_j^k = e^{k\Delta t \mu_j} e^{ik\Delta t \omega_j}$. Using this we can write

$$\hat{x}_{t+k}^j = \bar{K}_j^k x_t^j + \sum_{l=1}^{k} \bar{K}_j^{k-l} c_{t+l-1}^j$$

Now applying the chain rule, we get derivatives of the loss with respect to $x_t^j$ and $c_{t+l-1}^j$:

$$\implies \frac{\partial \mathcal{L}_k}{\partial x_t^j} = \frac{\partial \hat{x}_{t+k}^j}{\partial x_t^j} \frac{\partial \mathcal{L}_k}{\partial \hat{x}_{t+k}^j} = \bar{K}_j^k \frac{\partial \mathcal{L}_k}{\partial \hat{x}_{t+k}^j} = e^{k\Delta t \mu_j} e^{ik\Delta t \omega_j} \frac{\partial \mathcal{L}_k}{\partial \hat{x}_{t+k}^j}$$

$$\implies \frac{\partial \mathcal{L}_k}{\partial c_{t+l-1}^j} = \frac{\partial \hat{x}_{t+k}^j}{\partial c_{t+l-1}^j} \frac{\partial \mathcal{L}_k}{\partial \hat{x}_{t+k}^j} = \bar{K}_j^{k-l} \frac{\partial \mathcal{L}_k}{\partial \hat{x}_{t+k}^j} = e^{(k-l)\Delta t \mu_j} e^{i(k-l)\Delta t \omega_j} \frac{\partial \mathcal{L}_k}{\partial \hat{x}_{t+k}^j}$$

As $|e^{i\theta}| = 1$, we get the result of partial derivatives given in Theorem 3.1. $\square$

## B  ADDITIONAL DETAILS ON THE DERIVATIONS

We first provide additional steps for the derivation of Eq. (5). As $\hat{x}_{t+1} = \bar{K}x_t + \bar{L}u_t$, we can unroll it to get future predictions up to $\tau$ time steps that is:

$$\hat{x}_{t+1} = \bar{K}x_t + \bar{L}u_t$$
$$\hat{x}_{t+2} = \bar{K}\hat{x}_{t+1} + \bar{L}u_{t+1} = \bar{K}(\bar{K}x_t + \bar{L}u_t) + \bar{L}u_{t+1} = \bar{K}^2 x_t + \bar{K}\bar{L}u_t + \bar{L}u_{t+1}$$
$$\hat{x}_{t+3} = \bar{K}\hat{x}_{t+2} + \bar{L}u_{t+2} = \bar{K}(\bar{K}^2 x_t + \bar{K}\bar{L}u_t + \bar{L}u_{t+1}) + \bar{L}u_{t+2} = \bar{K}^3 x_t + \bar{K}^2 \bar{L}u_t + \bar{K}\bar{L}u_{t+1} + \bar{L}u_{t+2}$$
$$\cdots$$
$$\hat{x}_{t+\tau} = \bar{K}^\tau x_t + \bar{K}^{\tau-1}\bar{L}u_t + \bar{K}^{\tau-2}\bar{L}u_{t+1} + \cdots + \bar{L}u_{t+\tau-1}$$

Writing the above equations in a matrix form and using $c_{t+k} = \bar{L}u_{t+k}$ we get Eq. (5). Now to derive Eq. (7), we use the property of matrix exponential of diagonal matrices. In particular, we have $\bar{K} = \text{diag}(\bar{\lambda}_1, \ldots, \bar{\lambda}_m)$ implies that $\bar{K}^\tau = \text{diag}(\bar{\lambda}_1^\tau, \ldots, \bar{\lambda}_m^\tau)$. Using the notations from Section 3.2, we get that for $i$-th index of predicted vectors $\hat{x}_{t+k}$ s the following equations hold:

$$\hat{x}_{t+1}^i = \bar{\lambda}_i x_t^i + c_t^i$$
$$\hat{x}_{t+2}^i = \bar{\lambda}_i^2 x_t^i + \bar{\lambda}_i c_t^i + c_{t+1}^i$$
$$\cdots$$
$$\hat{x}_{t+\tau}^i = \bar{\lambda}_i^\tau x_t^i + \bar{\lambda}_i^{\tau-1} c_t^i + \bar{\lambda}_i^{\tau-2} c_{t+1}^i + \cdots + c_{t+\tau-1}^i$$

The above equations can be denoted by an expression using the circular convolution operator as written in Eq. (7).

## C    PROPOSED MODIFIED DYNAMICS MODEL IN RL AND PLANNING

### C.1    REINFORCEMENT LEARNING

The RL setting can be formalized as a Markov Decision Process (MDP), where an agent learns to make decisions by interacting with an environment. The agent's goal is to maximize a cumulative reward signal, where a *reward function* $r(s, a)$ assigns a scalar value to each state-action pair $(s, a)$. The agent's *policy*, denoted by $\pi(a|s)$, defines the probability distribution over actions at a given state. The agent's objective is to learn a policy that maximizes the expected discounted sum of rewards over time, also known as the return: $J(\pi) = \mathbb{E}_\pi \left[ \sum_{t=0}^{\infty} \gamma^t r(s_t, a_t) \right]$ where $\gamma \in [0, 1]$ is the *discount factor* and $s_t$ and $a_t$ are the state and action at time step $t$, respectively. The agent's expected cumulative reward can also be represented by the Value function, which is defined as the expected cumulative reward starting from a given state and following a given policy.

*Value-based* RL algorithms have two steps of policy evaluation and policy improvement. For example, *Q-learning* estimates the optimal action-value function, denoted by $Q^*(s, a)$, that represents the maximum expected cumulative reward achievable by taking action $a$ in state $s$ and improves the policy by setting it to $arg \max_a Q^*(s, a)$ (Mnih et al., 2013; 2015). A *policy-based* RL algorithm directly optimizes the policy to maximize the expected return. *Policy gradient* methods update the policy by adjusting the policy function's parameters by estimating the gradient of $J$. *Actor-critic* methods combine Q-learning and policy gradient ones, such that the Q-network (critic) learns the action value under the current policy, and the policy network (actor) tries to maximize it Haarnoja et al. (2018); Lillicrap et al. (2016); Schulman et al. (2017).

### C.2    FORWARD DYNAMICS MODELING IN RL

Dynamics models have been instrumental in attaining impressive results on various tasks such as Atari (Schrittwieser et al., 2020; Hafner et al., 2020) and continuous control (Hafner et al., 2020; Jiang et al., 2020; Sikchi et al., 2021; Lowrey et al., 2019). By building a model of environment dynamics, *model-based RL* can generate trajectories used for training the RL algorithm. This can significantly reduce the sample complexity in comparison to model-free techniques. However, in model-based RL, inaccurate long-term predictions can generate poor trajectory rollouts and lead to incorrect expected return calculations, resulting in misleading policy updates. Forward dynamics modeling has also been successfully applied to *model-free RL* to improve the sample efficiency of the existing model-free algorithms. These methods use it to design self-supervised auxiliary losses for representation learning using consistency in forward dynamics in the latent and the observation space (Jaderberg et al., 2017; Schwarzer et al., 2020; Srinivas et al., 2020).

### C.3    MODEL-BASED PLANNING

*Model Predictive Control* (MPC) is a control strategy that uses a dynamics model $s_{t+1} = f(s_t, a_t)$ to plan for a sequence of actions $a_t, a_{t+1}, \ldots, a_{t+\tau-1}$ that maximizes the expected return over a finite horizon. The optimization problem is:

$$\arg \max_{a_{t:t+\tau-1}} \mathbb{E} \left[ \sum_{i=t}^{t+\tau-1} \gamma^i r(s_i, a_i) \right] \tag{8}$$

where $\gamma$ is typically set to 1, that is, there is no discounting. Heuristic population-based methods, such as a cross entropy method Rubinstein & Kroese (2004) are often used for dynamic planning. These methods perform a local trajectory optimization problem, corresponding to the optimization of a cost function up to a certain number of time steps in the future (Williams et al., 2015; 2018; Okada & Taniguchi, 2020). In contrast to Q-learning, this is myopic and can only plan up to a certain horizon, which is predetermined by the algorithm.

One can also combine MPC with RL to approximate long-term returns of trajectories that can be calculated by bootstrapping the value function of the terminal state. In particular, methods like *TD-MPC* (Hansen et al., 2022) and *LOOP* (?) combine value learning with planning using MPC. The learned value function is used to bootstrap the trajectories that are used for planning using MPC. Additionally, they learn a policy using the *Deep Deterministic Policy Gradient* (Lillicrap et al., 2016) or *Soft Actor-Critic (SAC)* (Haarnoja et al., 2018) to augment the planning algorithm with good

proposal trajectories. Alternative search method such as Monte Carlo Tree search (Coulom, 2006) is used for planning in discrete action spaces. The performance of the model-based planning method heavily relies on the accuracy of the learned model.

## C.4 KOOPMAN TDMPC

TDMPC uses an MLP to design a Task-oriented Latent Dynamica (TOLD) model, which learns to predict the latent representations of the future time steps. The learnt model can then be used for planning. To stabilize training TOLD, the weights of a *target encoder network* $g_{\theta^-}$, are updated with the exponential moving average of the *online network* $g_\theta$.

Since dynamics modeling is no longer the only objective, in addition to the latent consistency of Eq. (6), the latent $x^t$ is also used to predict the reward and $Q$-function, which takes the latent representations as input. Hence, the dynamics model learns to jointly minimize the following:

$$\mathcal{L}_{\text{TOLD}} = c_1 \sum_{k=1}^{\tau} \lambda^k \|\hat{x}_{t+k} - g_{\theta^-}(s_{t+k})\|_2^2 + c_2 \sum_{k=0}^{\tau} \lambda^k \|R_\theta(\hat{x}_{t+k}, a_{t+k}) - r_{t+k}\|_2^2$$

$$+ c_3 \sum_{k=0}^{\tau} \lambda^k \|Q_\theta(\hat{x}_{t+k}, a_{t+k}) - y_{t+k}\|_2^2 \tag{9}$$

where $R_\theta, \pi_\theta, Q_\theta$ are respectively reward, policy and Q prediction networks, where $y_{t+k} = r_{t+k} + Q_{\theta^-}(\hat{x}_{t+k+1}, \pi_\theta(\hat{x}_{t+k}))$ is the 1-step bootstrapped TD target. Moreover, the policy network is learned from the latent representation by maximizing the Q value at each time step. This additional policy network helps to provide the next action for the TD target and a heuristic for planning. For details on the planning algorithm and its implementation, see Hansen et al. (2022).

## C.5 KOOPMAN SELF-PREDICTIVE REPRESENTATIONS

To avoid overconstraining the latent space using the dynamics model, SPR uses *projection heads*. Similarly, we encode the Koopman observables using a projection layer above the representations used for Q-learning. Using $s_t$ for the input pixel space, the representation space for the Q-learning is given by $z_t = e_\theta(s_t)$ where $e_\theta$ is a CNN. The Koopman observables $x_t$ is produced by encoding $z_t$ using a projector $p_\theta$ such that $x_t = p_\theta(z_t)$. Moreover, it can be decoded into $z_t$ using the decoder $d_\theta(\hat{x}_t)$. Loss functions are the prediction loss Eq. (6) and TD-error

$$\mathcal{L}_{\text{SSL}} = \sum_{k=1}^{\tau} \|d_\theta(\hat{x}_{t+k}) - e_{\theta^-}(s_{t+k})\|_2^2 + \|Q_\theta(z_t, a_t) - (r_t + Q_{\theta^-}(e_{\theta^-}(s_{t+1}), a_{t+1}))\|_2^2. \tag{10}$$

Here, $a_{t+1}$ is sampled from the policy $\pi$. The policy is learned from the representations using Soft Actor-Critic (SAC; Haarnoja et al., 2018). As opposed to SPR, we do not use moving averages for the target encoder parameters and simply stop gradients through the target encoders and denote it as $e_{\theta^-}$. Moreover, we drop the consistency term in the Koopman space in Eq. (10) as we empirically observed adding consistency both in Koopman observable ($x$), and Q-learning space ($z$) promotes collapse and makes training unstable, resulting in a higher variance in the model's performance.

## D COMPARISON WITH DSSM

While our proposed Koopman model may look similar to a DSSM Gu et al. (2022a); Gupta et al. (2022); Mehta et al. (2022), there are four major differences. First, our model is specifically derived for dynamics modeling with control input using Koopman theory and not for sequence or time series modeling. Our motivation is to make the dynamics and gradients through time stable, whereas, for DSSMs, it is to approximate the 1D convolution kernels derived from HiPPO Gu et al. (2020) theory. Second, a DSSM gives a way to design a cell that is combined with non-linearity and layered to get a non-linear sequence model. In contrast, our Koopman-based model shows that simple linear latent dynamics can be sufficient to model complex non-linear dynamical systems with control. Third, DSSMs never explicitly calculate the latent states and even ignore the starting state. Our model works in the state space, where the starting state is crucial to backpropagate gradients through the state encoder. Fourth, a DSSM learns structured convolution kernels for 1D to 1D signal mapping so that

for higher dimensional input, it has multiple latent dynamics models running under the hood, which are implemented as convolutions. In contrast to this, our model runs a single linear dynamics model for any dimensional input.

# E  DETAILED RESULTS FROM THE MAIN TEXT

In this section, we provide the numerical values in Table 1 and Table 2 of the results used in Fig. 3. Table 3 provides the numerical values for training speed used in Fig. 4. In Fig. 7 and Fig. 8, we further provide the environment-wise returns evaluated after every 5000 training steps for the planning and RL experiments used in Fig. 5 and Fig. 6.

Table 1: Forward state prediction error in Offline Reinforcement Learning environments. The reported number is the mean square error of the state prediction over a **horizon of 100 environment steps**. The results include standard deviation over 3 runs.

| Offline Datasets | State Prediction Error ($\downarrow$) | | | | |
| | MLP | Transformer | GRU | DSSM | Koopman |
| --- | --- | --- | --- | --- | --- |
| hopper-expert-v2 | $0.500 \pm 0.080$ | $0.060 \pm 0.007$ | $\mathbf{0.006 \pm 0.001}$ | $0.030 \pm 0.008$ | $0.007 \pm 0.000$ |
| hopper-medium-expert-v2 | $0.550 \pm 0.120$ | $0.063 \pm 0.009$ | $\mathbf{0.011 \pm 0.004}$ | $0.050 \pm 0.008$ | $0.012 \pm 0.004$ |
| hopper-full-replay-v2 | $0.880 \pm 0.150$ | $0.240 \pm 0.070$ | $\mathbf{0.050 \pm 0.005}$ | $0.180 \pm 0.050$ | $0.070 \pm 0.004$ |
| walker2d-expert-v2 | $0.300 \pm 0.060$ | $0.130 \pm 0.060$ | $0.100 \pm 0.005$ | $0.140 \pm 0.020$ | $\mathbf{0.090 \pm 0.008}$ |
| walker2d-medium-expert-v2 | $1.320 \pm 0.210$ | $0.310 \pm 0.070$ | $\mathbf{0.190 \pm 0.050}$ | $0.320 \pm 0.070$ | $0.220 \pm 0.040$ |
| walker2d-full-replay-v2 | $4.500 \pm 2.100$ | $3.300 \pm 0.350$ | $\mathbf{0.850 \pm 0.080}$ | $1.200 \pm 0.090$ | $0.910 \pm 0.050$ |
| halfcheetah-expert-v2 | $2.860 \pm 1.100$ | $0.840 \pm 0.160$ | $\mathbf{0.280 \pm 0.050}$ | $0.520 \pm 0.050$ | $0.300 \pm 0.060$ |
| halfcheetah-medium-expert-v2 | $0.590 \pm 0.090$ | $0.540 \pm 0.071$ | $\mathbf{0.501 \pm 0.031}$ | $0.910 \pm 0.040$ | $0.568 \pm 0.040$ |
| halfcheetah-full-replay-v2 | $0.850 \pm 0.200$ | $0.930 \pm 0.140$ | $0.750 \pm 0.030$ | $0.950 \pm 0.060$ | $\mathbf{0.700 \pm 0.040}$ |

Table 2: Forward reward prediction error in Offline Reinforcement Learning environments. The reported number is the mean square error of the reward prediction over a **horizon of 100 environment steps**. The results include standard deviation over 3 runs.

| Offline Datasets | Reward Prediction Error ($\downarrow$) | | | | |
| | MLP | Transformer | GRU | DSSM | Koopman |
| --- | --- | --- | --- | --- | --- |
| hopper-expert-v2 | $0.080 \pm 0.005$ | $0.030 \pm 0.002$ | $\mathbf{0.001 \pm 0.000}$ | $0.002 \pm 0.000$ | $\mathbf{0.001 \pm 0.000}$ |
| hopper-medium-expert-v2 | $0.260 \pm 0.050$ | $0.046 \pm 0.008$ | $\mathbf{0.001 \pm 0.000}$ | $0.005 \pm 0.000$ | $0.002 \pm 0.000$ |
| hopper-full-replay-v2 | $0.400 \pm 0.100$ | $0.110 \pm 0.040$ | $\mathbf{0.009 \pm 0.005}$ | $0.020 \pm 0.004$ | $0.012 \pm 0.003$ |
| walker2d-expert-v2 | $0.015 \pm 0.003$ | $0.004 \pm 0.001$ | $0.004 \pm 0.001$ | $0.006 \pm 0.001$ | $\mathbf{0.003 \pm 0.000}$ |
| walker2d-medium-expert-v2 | $0.190 \pm 0.030$ | $0.032 \pm 0.009$ | $\mathbf{0.009 \pm 0.003}$ | $0.015 \pm 0.003$ | $0.011 \pm 0.002$ |
| walker2d-full-replay-v2 | $0.520 \pm 0.080$ | $0.220 \pm 0.020$ | $\mathbf{0.080 \pm 0.015}$ | $0.091 \pm 0.005$ | $0.090 \pm 0.008$ |
| halfcheetah-expert-v2 | $0.160 \pm 0.050$ | $0.070 \pm 0.005$ | $0.045 \pm 0.004$ | $0.120 \pm 0.020$ | $\mathbf{0.040 \pm 0.006}$ |
| halfcheetah-medium-expert-v2 | $0.230 \pm 0.050$ | $0.120 \pm 0.070$ | $\mathbf{0.050 \pm 0.007}$ | $0.150 \pm 0.007$ | $0.060 \pm 0.005$ |
| halfcheetah-full-replay-v2 | $0.170 \pm 0.045$ | $0.110 \pm 0.052$ | $0.105 \pm 0.009$ | $0.205 \pm 0.011$ | $\mathbf{0.091 \pm 0.006}$ |

Table 3: Training speed (iterations/second) for a state prediction task using an MLP, a Transformer, and a diagonal-Koopman dynamics model on halfcheetah-expert-v2. Each iteration consists of one gradient update of the entire model.

| Dynamics Model Type | Training sequence length | | | | |
| | 10 | 50 | 100 | 200 | 500 |
| --- | --- | --- | --- | --- | --- |
| MLP | 478.00 | 115.54 | 55.85 | 28.02 | 11.20 |
| Transformer | 418.00 | 120.20 | 73.34 | 34.48 | 14.92 |
| GRU | 420.00 | 101.20 | 42.02 | 17.40 | 8.91 |
| DSSM | 530.32 | 148.50 | 76.10 | 37.12 | 17.20 |
| Koopman | **565.43** | **160.76** | **86.49** | **59.65** | **24.56** |

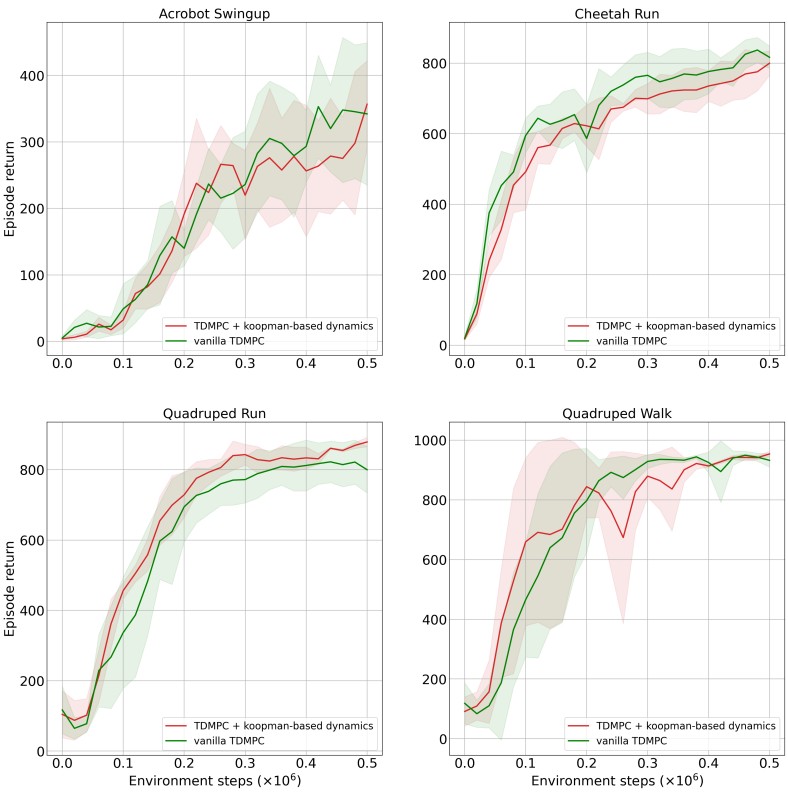

Figure 7: Comparison of our Koopman-based dynamics model (with a horizon of 20) with an MLP-based dynamics model of vanilla TD-MPC Hansen et al. (2022). The results are shown with 5 random seeds with the standard deviation shown using the shaded regions.

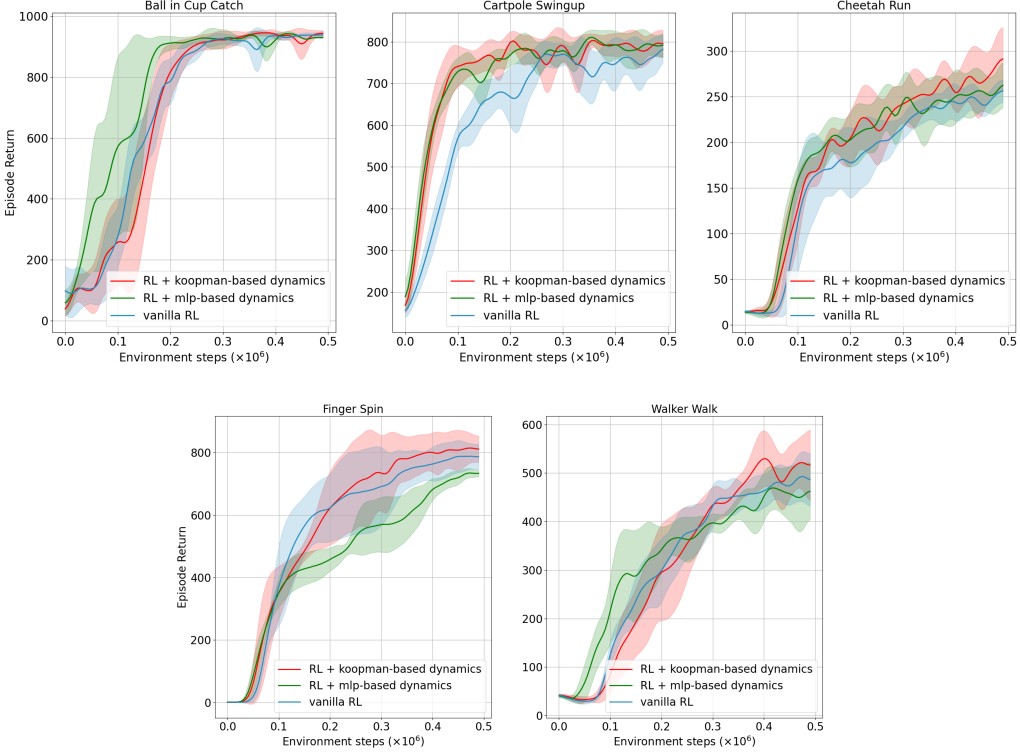

Figure 8: Comparison of vanilla RL (SAC Haarnoja et al. (2018)) and its integration with an MLP-based (SPR Schwarzer et al. (2020)) and a Koopman-based dynamics model for incorporating self-predictive representations in the DeepMind Control Suite. Shaded regions represent the standard deviation.

## F  MORE DETAILS ON THE BASELINE DYNAMICS MODELS

For all the experiments, we used a model with around $\sim 500k$ trainable parameters to ensure that the improved compute efficiency during training is not due to the model having fewer parameters. To incorporate the causal transformer architecture with that many parameters, we reduced the embedding dimension of both the state and action embedding to $64$. We further reduced the number of heads to $4$ to design a causal Transformer model that matches the size of our Koopman model. For the GRU and MLP-based model, we keep these dimensions the same as the Koopman one. We use a hidden dimension size of 64 for the GRU based-model in order to keep the parameters similar. Moreover, all our experiments in Table 3 use one Nvidia A100 GPU for training.

## G  ABLATION STUDY ON INITIALIZATION

We now perform an ablation study on the different possible initialization methods presented in Section 3.2.2. We report the results for `halfcheetah-expert-v2`, `walker-2d-expert-v2` and `hopper-expert-v2`, in Table 4, 5 and 6 respectively. We see a constant value of $-0.2$ for $\mu_i$, and increasing the order of frequency for $\omega_i$ gives a consistent performance boost over different environments.

Table 4: Ablation on initialization of the eigenvalues of the diagonal Koopman matrix: $\lambda_i = \mu_i + j\omega_i$ for `halfcheetah-expert-v2`. The results are over three seeds.

| $\mu_i$ | $\omega_i$ | State Prediction Error ($\downarrow$) | Reward Prediction Error ($\downarrow$) |
|---|---|---|---|
| constant | increasing | **0.300 $\pm$ 0.060** | **0.040 $\pm$ 0.006** |
| constant | random | 0.340 $\pm$ 0.047 | 0.041 $\pm$ 0.005 |
| learnable | increasing | 0.320 $\pm$ 0.024 | 0.048 $\pm$ 0.003 |
| learnable | random | 0.360 $\pm$ 0.040 | 0.042 $\pm$ 0.007 |

Table 5: Ablation on initialization of the eigenvalues of the diagonal Koopman matrix: $\lambda_i = \mu_i + j\omega_i$ for `walker-2d-expert-v2`. The results are over three seeds.

| $\mu_i$ | $\omega_i$ | State Prediction Error ($\downarrow$) | Reward Prediction Error ($\downarrow$) |
|---|---|---|---|
| constant | increasing | **0.090 $\pm$ 0.008** | **0.003 $\pm$ 0.000** |
| constant | random | 0.104 $\pm$ 0.008 | 0.004 $\pm$ 0.001 |
| learnable | increasing | 0.102 $\pm$ 0.003 | 0.004 $\pm$ 0.000 |
| learnable | random | 0.096 $\pm$ 0.007 | 0.003 $\pm$ 0.002 |

Table 6: Ablation on initialization of the eigenvalues of the diagonal koopman matrix: $\lambda_i = \mu_i + j\omega_i$ for `hopper-expert-v2`. The results are over three seeds.

| $\mu_i$ | $\omega_i$ | State Prediction Error ($\downarrow$) | Reward Prediction Error ($\downarrow$) |
|---|---|---|---|
| constant | increasing | **0.007 $\pm$ 0.000** | **0.001 $\pm$ 0.000** |
| constant | random | 0.010 $\pm$ 0.007 | 0.014 $\pm$ 0.005 |
| learnable | increasing | 0.008 $\pm$ 0.006 | 0.002 $\pm$ 0.004 |
| learnable | random | 0.008 $\pm$ 0.004 | 0.003 $\pm$ 0.001 |

## H   ABLATION STUDY ON HORIZON FOR PLANNING AND MODEL-FREE RL

We investigate the role of horizon (or rollout length) in both model-based planning (Section 4.2.1) and model-free RL (Section 4.2.2) for our Koopman-based dynamics model. The corresponding results are available in Fig. 9 and Fig. 10. We observe that for model-based planning, the performance improves with longer horizon. However, for model-free RL, the performance is higher with a prediction horizon of 5 and deteriorates for 20. This phenomenon is consistent with the results reported in Figure 4 of Schwarzer et al. (2020).

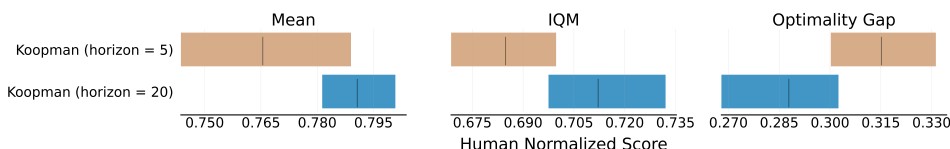

Figure 9: Ablations of our Koopman-based dynamics model for horizon lengths of 5 and 20. The results are over 5 random seeds for each environment. Higher Mean & IQM and lower Optimality Gap is better.

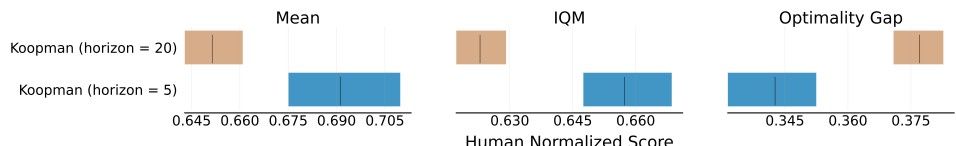

Figure 10: Ablations of our Koopman-based dynamics model for horizon lengths of 5 and 20 for incorporating self-predictive representations with vanilla SAC Haarnoja et al. (2018) in the DeepMind Control Suite. The results are over 5 random seeds for each environment. Higher Mean & IQM and lower Optimality Gap is better.

## I  ABLATION ON THE DIMENSIONALITY OF LINEAR LATENT SPACE

We also ablate the dimensionality of the Koopman state observables: $m$ in Table 7 and show that a $512$ is enough for most of the environments, as the performance saturates there.

Table 7: Forward state prediction error in Offline Reinforcement Learning environments. The reported number is the mean square error of the state prediction in `halfcheetah-expert-v2` over a horizon of 100 environment steps. We observe a general trend of a decrease in the error values with an increase in the state embedding dimension which is aligned with the Koopman theory. We have selected an embedding dimension of 512 for all our experiments as it has the lowest error value.

| State Embedding Dim. ($m$) | State Prediction Error ($\downarrow$) |
|---|---|
| 128 | 0.0097 |
| 256 | 0.0081 |
| 512 | **0.0073** |
| 1024 | 0.0075 |

## J  KOOPMAN DYNAMICS MODEL IN JAX

We provide the Jax code snippet to efficiently implement our Koopman-based dynamics model. Our code can be found in https://github.com/arnab39/koopman-dynamica.

---

**Algorithm 1** Diagonal Koopman Dynamics model

---

```python
import jax
import jax.numpy as jnp
from jax.nn.initializers import normal
from flax import linen as nn

# Main class to implement Diagonal Koopman Dynamics model
class BatchedDiagonalKoopmanDynamicsModel(nn.Module):
    state_dim: int
    action_emb_dim: int
    real_init_type: str = 'constant'
    real_init_value: float = -0.5
    im_init_type: str = 'increasing_freq'
    activations: Callable[[jnp.ndarray], jnp.ndarray] = nn.relu

    def init_koopman_imaginary_params(self):
        if self.im_init_type == 'increasing_freq':
            self.K_im = self.param(
            "K_im", increasing_im_init(), (self.state_dim // 4,)
            )
        elif self.im_init_type == 'random':
            self.K_im = self.param(
            "K_im", random_im_init(), (self.state_dim // 4,)
            )
        else:
            raise ValueError

    def init_koopman_real_params(self):
        if self.real_init_type == 'constant':
            self.K_real = self.real_init_value
        elif self.real_init_type == 'learnable':
            self.K_r = self.param(
            "K_real", nn.initializers.ones, (self.state_dim // 4,)
            ) * self.real_init_value
            self.K_real = jnp.clip(self.K_r, -0.4, -0.1)
        else:
            raise ValueError

    def setup(self):
        # model parameters
        self.init_koopman_real_params()
        self.init_koopman_imaginary_params()
        self.action_encoder = MLP(
        [128, self.action_emb_dim * 2],
        activations=self.activations
        )
        self.L = self.param(
        "L", normal(0.1), (self.action_emb_dim, self.state_dim // 2, 2)
        )

        # Step parameter
        self.step = jnp.exp(self.param("log_step", log_step_init(), (1,)))

        self.K_complex = jnp.concatenate(
            [self.K_real + 1j * self.K_im,
            self.K_real - 1j * self.K_im],
            axis=-1
        )
        self.L_complex = self.L[:, :, 0] + 1j * self.L[:, :, 1]
        self.K_dis, self.L_dis = discretize(
            self.K_complex, self.L_complex, self.step
        )

    def __call__(self, actions, start_state_rep):
        if len(actions.shape) == 2:
            return koopman_forward_single(
                self.K_dis, self.L_dis,
                self.action_encoder(actions), start_state_rep
            )
        else:
            Vand_K = jnp.vander(
                self.K_dis, actions.shape[1] + 1, increasing=True
            )
            return koopman_forward(
                Vand_K, self.L_dis,
                self.action_encoder(actions), start_state_rep
            )
```

---

---

**Algorithm 2** Helper functions for Diagonal Koopman Dynamics model

---

```
def causal_convolution(u, K):
    assert K.shape[0] == u.shape[0]
    ud = jnp.fft.fft(jnp.pad(u, (0, K.shape[0])))
    Kd = jnp.fft.fft(jnp.pad(K, (0, u.shape[0])))
    out = ud * Kd
    return jnp.fft.ifft(out)[: u.shape[0]]

def compute_measurement_block(Ker, action_emb, init_state):
    return causal_convolution(action_emb, Ker[:-1]) + Ker[1:] * init_state

def compute_measurement(Ker, action_emb, init_state):
    return jax.vmap(
            compute_measurement_block,
            in_axes=(0, 1, 0),
            out_axes=1
        )(Ker, action_emb, init_state)

def discretize(K, L, step):
    return jnp.exp(step*K), (jnp.exp(step*K)-1)/K * L

def log_step_init(dt_min=0.001, dt_max=0.1):
    def init(key, shape):
        return jax.random.uniform(key, shape) * (
            jnp.log(dt_max) - jnp.log(dt_min)
        ) + jnp.log(dt_min)
    return init

def increasing_im_init():
    def init(key, shape):
        return jnp.pi * jnp.arange(shape[0])
    return init

def random_im_init():
    def init(key, shape):
        return jax.random.uniform(key, shape)
    return init

@jit
def koopman_forward(Vand_K, L, actions_emb, start_state_rep):
    action_emb_dim = actions_emb.shape[-1]
    state_dim = start_state_rep.shape[-1]
    actions_emb_complex = actions_emb[:, :, :action_emb_dim//2] \
                            + 1j * actions_emb[:, :, action_emb_dim//2:]
    start_state_rep_complex = start_state_rep[:, :state_dim // 2] \
                                + 1j * start_state_rep[:, state_dim // 2:]

    predicted_state_rep_complex = jax.vmap(
        compute_measurement,
        in_axes=(None, 0, 0),
        out_axes=0
    )(Vand_K, actions_emb_complex @ L, start_state_rep_complex)

    predicted_state_rep = jnp.concatenate([
        jnp.real(predicted_state_rep_complex),
        jnp.imag(predicted_state_rep_complex)
    ], -1)
    return predicted_state_rep

@jit
def koopman_forward_single(K, L, action_emb, start_state_rep):
    action_emb_dim = action_emb.shape[-1]
    state_dim = start_state_rep.shape[-1]
    action_emb_complex = action_emb[:, :action_emb_dim//2] \
                            + 1j * action_emb[:, action_emb_dim//2:]
    start_state_rep_complex = start_state_rep[:, :state_dim // 2] \
                                + 1j * start_state_rep[:, state_dim // 2:]
    predicted_state_rep_complex = K[None, :] * start_state_rep_complex + \
                                    .action_emb_complex @ L
    predicted_state_rep = jnp.concatenate([
        jnp.real(predicted_state_rep_complex),
        jnp.imag(predicted_state_rep_complex)
    ], -1)
    return predicted_state_rep
```

---

