# OpenReview forum: "Efficient Dynamics Modeling in Interactive Environments with Koopman Theory"
_ICLR.cc/2024/Conference — ICLR 2024 poster_

### Official Review · Reviewer_tVSX · 2023-10-30

**Soundness:** 3 good
**Presentation:** 3 good
**Contribution:** 3 good
**Rating:** 8
**Confidence:** 4

**Summary:**

The authors propose a new state-space model for dynamical systems modelling based on the Koopman operator. The authors approximate the Koopman operator in such a way that they can leverage time convolution operators and compute multiple-step predictions without resorting to autoregressive filtering. This seems to reduce training time and improve learning algorithm stability. This is because the gradient growth can be directly controlled due to the structure of the proposed model. The authors also show that replacing a native model in some model-based algorithms with the proposed one improves the algorithm's performance.

**Strengths:**

1. The idea of creating a linear approximation of the nonlinear dynamical system through a Koopman framework is very interesting. This is because we can use a well-developed linear systems theory to derive scalable algorithms.
1. The theoretical result on gradient growth is valuable for practitioners, who are not well acquainted with dynamical systems theory and stability theory
1. The authors demonstrate on some numerical examples, that the generative model is efficient for learning a planning controller as well as a policy.

**Weaknesses:**

1. Contributions are not well articulated in the introduction. It is not clear how this work is different from other Koopman operator-based RL methods. For example, I find the tools to compute long-term predictions quite interesting and a bit more detail on this in the introduction would strengthen the paper.
1. Ablation on sequence length for RL tasks is not provided, i.e., do we get an improvement from the model being able to predict several steps ahead? This is an important ablation since it was shown by Janner et al that using SAC as a policy learning algorithm does not benefit from rollouts longer than a one-time step. One potential reason for this is the algorithm itself that learns only from one-step-ahead predictions. Therefore, this raises the question of whether we actually need a model capable of long-term predictions. I do not see a discussion on this subject.
1. Ablation with Hippo model as a backbone for offline RL is missing. The authors mention the relation to the Koopman framework and it would be interesting to see if the proposed model offers an improvement over HIPPO
1. The presentation needs to be sharpened a bit to highlight the main contribution - the effect of the novel Koopman model on RL in comparison to other generative models. I think the authors can pivot to argue that their Koopman model should be considered as the base model for most of the RL problems (offline RL, planning, model-based RL etc). But this a matter of choice.

I will consider my score further in the review process

**Questions:**

1. “In this work, we leverage techniques and perspectives from Koopman theory (Koopman, 1931; Koopman & Neumann, 1932; Brunton et al., 2021) to address this key problem in long-range dynamics modeling of interactive environments.” I feel that Mauroy et al 2020 would be a better reference for modern Koopman theory and recent developments
1. “and thus an invariant subspace $G \subset F$ is often used, so that $Kg \in G$. $G$ is spanned by a finite set of observables $g_1$, . . . , $g_m$, where often we assume $m \gg n$.”\
This is slightly confusing. Finding an invariant subspace *is not a tractable problem*, while this phrasing implies that we can pick an invariant subspace. Please rephrase.
1. Page 3. “one could show that the Koopman operator is bilinearized (Brunton et al., 2021)”
If I understood correctly Brunton et al bilinearization is possible only if there is a finite number of eingenfunctions (Theorem 6.1). I am not sure if this implies isomorphism to linear dynamics, but in any case, this is an important restriction worth mentioning

1. “An alternate approach to simplifying the Koopman operator with a control signal assumes”. “Simplifying” is a confusing word in this context. I recommend using “approximating”, which would rule out equivalence of representations.

1. “Fig. 4 empirically verifies that our proposed Koopman dynamics model is around twice as fast as an MLP, GRU or Transformer based dynamics model in learning from longer trajectories” It is hard to verify this claim based on the figure, I suggest referencing Table 3 in the appendix and provide the number 1.7 instead of “around twice as fast”

1. There seems to be a small clash of notation, where $\overline \lambda_j$ is used to describe the entries of the diagonal Koopman matrix $\overline K$, and also the diagonal entries of the matrix $K$.

References:

Janner, Michael, et al. "When to trust your model: Model-based policy optimization." Advances in neural information processing systems 32 (2019).

Fujimoto, Scott, and Shixiang Shane Gu. "A minimalist approach to offline reinforcement learning." Advances in neural information processing systems 34 (2021): 20132-20145.

Mauroy, Alexandre, Y. Susuki, and I. Mezić. Koopman operator in systems and control. Berlin: Springer International Publishing, 2020.

---

> ### Author Response · Authors · 2023-11-21
> **Official response to Reviewer tVSX**
>
> We thank the reviewer for their useful feedback and for appreciating the strengths of our approach.
>
> Below, we address the reviewer’s concerns:
>  > Contributions are not well articulated in the introduction.
>  > The presentation needs to be sharpened a bit to highlight the main contribution - the effect of the novel Koopman model on RL in comparison to other generative models.
>
> Following the reviewer’s suggestion, we have now rewritten the contribution section in the revised manuscript to better highlight the main contributions of our work.
>
> > It is not clear how this work is different from other Koopman operator-based RL methods.
>
> We have further extended the discussion of related work on Koopman-based RL in Section 5, where we also contrast these with our approach.
>
> > Ablation with Hippo model as a backbone for offline RL is missing. The authors mention the relation to the Koopman framework and it would be interesting to see if the proposed model offers an improvement over HIPPO
>
> We thank the reviewer for suggesting a suitable baseline. In the revised submission, we added the DSSM/S4D [1,2] model results as a backbone for offline RL. Note that the authors of [1] have shown that the DSSM with certain initialization of diagonal parameters approximates the convolution kernel derived from the HiPPO framework.[3] Please refer to Figure 3/Table 1, 2 and Figure 4/Table 3 for the new results. Our model is faster than the DSSM baseline with a similar number of parameters while being better/competitive in terms of performance.
>
> >Ablation on sequence length for RL tasks is not provided, i.e., do we get an improvement [………..] Therefore, this raises the question of whether we actually need a model capable of long-term predictions. I do not see a discussion on this subject.
>
> We thank the reviewer for suggesting an ablation of sequence length for RL experiments. We provide those experiments for both RL and planning in Appendix H. Now, we address the reviewer’s concern regarding the use of long-term predictions.
> We would like to point out that the paper (MBPO) [4] the reviewer referred to is on model-based RL, which, as mentioned in the manuscript, is left for future work. In the future, we plan to expand our dynamics model to model stochastic transition dynamics while still ensuring parallel training. This will allow for easy integration of algorithms like MBPO that require uncertainty estimation with model ensembles.
>
> For model-based planning, we observed that the results improved with higher rollout length, i.e., a horizon of 20 compared to 5. We have included the ablation of length for model-based planning in Figure 9 in the revised manuscript.
> However, for model-free RL with dynamics modelling as an auxiliary task, we notice that the performance drops when we increase the sequence length from 5 to 20. This is consistent with the ablation on the horizon provided in Figure 4 (and Section 5) of the SPR [5], where the authors show that the performance starts dropping after a sequence length of 7. Notably, even with a sequence length of just 5 our Koopman-based method achieves better performance.
> $$$$
> We also thank the reviewer for all the suggestions provided as points 1-6 in the Questions section. We have incorporated them all in the revised manuscript.
>
> $$$$
>
> [1] Gu, Albert, et al. "On the parameterization and initialization of diagonal state space models." Advances in Neural Information Processing Systems 35 (2022).
>
> [2] Gupta, Ankit, et al. "Diagonal state spaces are as effective as structured state spaces." Advances in Neural Information Processing Systems 35 (2022).
>
> [3] Gu, Albert, et al. "Hippo: Recurrent memory with optimal polynomial projections." Advances in neural information processing systems 33 (2020).
>
> [4] Janner, Michael, et al. "When to trust your model: Model-based policy optimization." Advances in neural information processing systems 32 (2019).
>
> [5] Schwarzer, Max, et al. "Data-Efficient Reinforcement Learning with Self-Predictive Representations." International Conference on Learning Representations. 2020.

---

> > ### Comment · Reviewer_tVSX · 2023-11-21
> > **Thank you for the response!**
> >
> > I want to thank the authors for their comprehensive responses to my questions and "weaknesses".
> >
> > I feel though that leaving Model-based RL for future work is a strange choice while presenting a model learning method. I guess for a Markovian fully observable case any Koopman model wouldn't add much. This could be specifically that there's no need for long-term predictions. If the authors have some preliminary experiments for this model-base case, which were not successful, I recommend including a discussion on these experiments. Just a few sentences would suffice.
> >
> > However, the new model could still be very useful in the POMDP case. For example, this could be the case in learning from pixels tasks, which authors consider. Perhaps, it's worth rephrasing "the model-free + SPR" subsection as an initial POMDP study. I leave it to the authors to decide.

---

> > > ### Author Response · Authors · 2023-11-22
> > > **Official comment to Reviewer tVSX response**
> > >
> > > We thank the reviewer for reconsidering the score and their new comments.
> > >
> > > We are indeed trying to integrate our model into model-based RL next. However, as state-of-the-art methods model stochastic dynamics auto-regressively, the current Koopman formulation can't be parallelized in that case. We want to retain the speed benefits while we integrate our model and hence are looking into stochastic Koopman dynamics to have a competitive baseline that can be parallelized during model training and roll-outs. We can add a proper discussion on this in the appendix. We can also provide some preliminary mode-based experiments in the final version of this paper if the reviewer feels that is necessary.

---

> > > > ### Comment · Reviewer_tVSX · 2023-11-22
> > > > **Thanks for the response**
> > > >
> > > > I will leave this up to you.

---

### Official Review · Reviewer_3syb · 2023-10-30

**Soundness:** 3 good
**Presentation:** 3 good
**Contribution:** 3 good
**Rating:** 5
**Confidence:** 3

**Summary:**

This paper introduces a new method for learning a Koopman embedding that gives decoupled linear dynamics in a latent space, opposed to nonlinear dynamics in the state space. This approach is also able to specify the eigenvalue spread of the resulting linear dynamics, stabilizing gradient evaluations through long chains.

**Strengths:**

1. There are many papers that propose using neural networks to find a Koopman embedding, this one is unique (to my knowledge) because it focuses on stabilizing the eigenvalues of the latent-space dynamics.
2. The included JAX code is very helpful for comparing the implementation to algorithm for a better understanding of what's going on.
3. The Koopman background section is clear and does a good job of defining all of the notation.
4. The argument for why a Koopman model is easier to parallelize over RNN's like GRU and LSTM is a great addition to the algorithm.
5. This approach was shown on a wide variety of robotics tasks from the DeepMind control suite (Ball in cup catch, cartpole, swingup, cheetah run, finger spin, and walker walk).

**Weaknesses:**

1. Figure 3 seems to undermine the utility of the presented Koopman approach, since the GRU is able to compete with it on all of the examples and metrics. Maybe some explanation in the figure caption explaining why this is expected would be appreciated.
2. Longer/better figure captions would be nice.
3. This seems like a marginal improvement over existing works cited. Some more comparisons between this approach and existing algorithms could better showcase the additions.

**Questions:**

N/A.

---

> ### Author Response · Authors · 2023-11-21
> **Official response to Reviewer 3syb**
>
> We thank the reviewer for their valuable feedback and for identifying the key strengths of our approach.
> Below, we address the reviewer’s concerns:
>
> > Figure 3 seems to undermine the utility of the presented Koopman approach, since the GRU is able to compete with it on all of the examples and metrics. Maybe some explanation in the figure caption explaining why this is expected would be appreciated.
> GRU-based dynamics modeling is indeed competitive with our approach and is widely used in model-based RL algorithms. However, as shown in Figure 4/Table 3, GRUs are significantly slower during training compared to our method. We aimed to build a simple latent linear model which can be competitive with existing ones while being several times faster and stable while training with a longer horizon. Following the reviewer’s suggestion, we have added more explanation to our captions in the revised manuscript.
>
> > Longer/better figure captions would be nice.
>
> We have added further explanation to the captions of Figures 3, 4, 5 and 6. We trust that these explanations will lead to a better understanding of the Figures presented.
>
> > This seems like a marginal improvement over existing works cited. Some more comparisons between this approach and existing algorithms could better showcase the additions.
>
> The main focus of our paper is on a latent linear dynamics model using Koopman theory so that we can significantly speed up the training. As shown in Figure 3/Table 1, 2 and Figure 4/Table 3, our model is significantly faster than existing baselines (which include MLP, Transformer, and GRU) with a similar number of parameters while being better/competitive in terms of performance. In the revised version, we additionally provide the results with the Diagonal State Space Model (DSSM)[1,2,3] for offline RL. The addition of DSSM as a baseline further strengthens our results since DSSMs are closely related to our model, as discussed in Appendix D. We demonstrate that while a DSSM-based dynamics model offers faster performance compared to other baseline approaches, it still lags behind our method in terms of speed. Moreover, it also falls short in state prediction accuracy when contrasted with our approach in the offline RL datasets.
> If the reviewer is suggesting the inclusion of additional baselines in our model-free RL and model-based planning experiments, we are certainly open to exploring additional relevant algorithms. We would appreciate any specific baseline recommendations that the reviewer had in mind. Please note that experiments provided in Sections 4.2.1 and 4.2.2 are to demonstrate that our method can be easily incorporated into existing RL and planning algorithms that use a dynamics model to enhance their performance
>
> $$$$
>
> [1] Gu, Albert, et al. "On the parameterization and initialization of diagonal state space models." Advances in Neural Information Processing Systems 35 (2022).
>
> [2] Gupta, Ankit, et al. "Diagonal state spaces are as effective as structured state spaces." Advances in Neural Information Processing Systems 35 (2022).
>
> [3] Mehta, Harsh, et al. "Long range language modeling via gated state spaces." arXiv preprint arXiv:2206.13947 (2022).

---

> > ### Author Response · Authors · 2023-11-22
> > **Gentle reminder: last day of rebuttal phase**
> >
> > Dear Reviewer 3syb,
> >
> > As we approach the conclusion of the rebuttal phase today, we kindly request an acknowledgement of our response. We believe we have addressed most of the reviewer's concerns.
> >
> > We appreciate your generally positive evaluation of our work and initial high score on soundness, presentation and contributions (3/4). It would be really valuable for us if the reviewer could point out any remaining concerns that may help them reconsider the final rating.
> >
> > Thank You,
> >
> > Authors

---

> > > ### Comment · Reviewer_3syb · 2023-11-23
> > >
> > > Thank you for the comments and answering my questions!

---

### Official Review · Reviewer_UaZL · 2023-10-31

**Soundness:** 3 good
**Presentation:** 3 good
**Contribution:** 3 good
**Rating:** 6
**Confidence:** 3

**Summary:**

In this paper, the authors introduce a diagonal Koopman-operator approach that helps to efficiently learn and quickly plan with dynamic models. The standard encoder used in Koopman-operator-based approaches is modified to output the complex eigenvectors of the latent linear model, whose estimation is then reduced to learning a diagonal matrix (together with the encoder/decoder weights). By careful initialization of the complex eigenvalues of the Koopman-matrix the authors can also protect against vanishing/exploding gradients. The diagonalization helps quite significantly to reduce the computation time needed for planning over long horizons, as the authors show in several dynamics-estimation and RL based tasks. The method also works competitively with several state-of-the-art approaches in dynamics modeling and RL applications.

**Strengths:**

The approach as far as I can tell is sound, and as the summary above indicates, the authors show significant improvement in terms of computation time / backpropagation stability that results from the diagonalization and the careful initialization. The method is also tested in several benchmark problems (RL and dynamics modeling) and the ablation studies in the appendix look comprehensive to me.

**Weaknesses:**

Sometimes the discussions in the paper are not very clear, for instance

* "While the same dynamics model can also be used for (III) model-based RL, we leave that direction for future work."
> Do you mean you would improve the model K further? or g_theta perhaps?

* Can your approach fail even if eigenvalues are initialized well? After correct initialization, will the real part of the eigenvalues always stay negative? Are there any other failure scenarios?

* "we apply our dynamics model in model-free RL, "
> What does it mean to apply a model in the model-free setting? Please explain not just in the appendix but also in the main text briefly. In general much of the discussion in the Appendix could be streamlined with the text. Please organize the appendix and reduce its scope, code should be put in github and linked, no need for printing your code in the appendix. Instead you could mention more failure cases as part of the ablation studies perhaps.

* Would be nice to have figures or a section explaining *what* you're solving, rather than how, e.g. inputs are in the pixel space, encoder is an MLP and the TD-MPC tries to solve X ... So some part of Section E from the appendix could go to the main text.

* It is not clear to me which network architecture you used for the encoder g_{\theta}, is it another MLP?

**Questions:**

* This is not a problem only concerning this work but in general applies to all 'Koopman-operator-inspired-approaches': the connection to the theory is very very loose and hence these methods could instead be simply called linear latent-dynamics modelling. Do you agree? If not, please mention how the Koopman-operator theory guides you analytically and intuitively.

* As a follow-up to the above: how does the encoder quality effect your results? How do you come up with the encoder structure? Can the theory guide you here?

* In the optimized Koopman matrix, are the eigenvalues sparse, or are they all nonzero/complex ? Can they be made sparse and is that interesting?

* Please mention the limitation to deterministic environments in the introduction. Is this a limitation that all Koopman-operator approaches in the literature face?

---

> ### Author Response · Authors · 2023-11-21
> **Official response to Reviewer UaZL (1/2)**
>
> We thank the reviewer for their positive assessment of our work and valuable suggestions.
>
> Below, we address the reviewer’s concerns:
> > "While the same dynamics model can also be used for (III) model-based RL, we leave that direction for future work." Do you mean you would improve the model K further? or g_theta perhaps?
>
> In the current submission, we have explored the integration of our Koopman-based model in model-based planning and model-free RL with dynamics modeling. Although our model can be applied to model-based RL, most state-of-the-art methods require stochastic dynamics modeling. As mentioned in Section 7, in the future, we plan to extend our framework to stochastic dynamics models and integrate it with model-based RL algorithms.
>
> > Can your approach fail even if eigenvalues are initialized well? After correct initialization, will the real part of the eigenvalues always stay negative? Are there any other failure scenarios?
>
> We thank the reviewer for raising an interesting question. We perform ablation studies of eigenvalue initialization in Appendix G (Tables 4, 5, and 6): a) constant $\mu_i$s, where we fix $\mu_i$s to be a real negative number throughout training and b) learnable $\mu_i$s, where each $\mu_i$ is bounded between two real negative numbers, such as [-0.3, -0.1]. In both cases, $\mu_i$s remains negative once initialized, and our theory suggests that as long as that stays negative, the training should be stable. Additionally, we have not seen any failure cases in training with these initialization settings.
>
> > "we apply our dynamics model in model-free RL, "
> What does it mean to apply a model in the model-free setting? [.....] Instead you could mention more failure cases as part of the ablation studies perhaps.
>
> We appreciate the reviewer’s suggestions.
> Current state-of-the-art model-free RL methods [1] utilize a dynamics model to improve the sample efficiency of existing model-free algorithms. These methods use a dynamics model to design a self-supervised auxiliary loss for representation learning by ensuring consistency in forward dynamics in the latent space. We have rewritten the first two sentences of Section 4.2.2 and have clarified  this.
> Regarding the code, as pointed out by Reviewer 3syb, having the main code snippet for the Koopman dynamics modeling part can ease the understanding of our algorithm for practitioners. As part of the revised submission, we have also added a link to an anonymous GitHub repository with the code base for the offline RL experiments with the baselines in Section J.
> Since we are constrained by the page limit, we have attempted to provide the most important experimental details in the main text and refer the readers to the appendix for additional details.
>
> >Would be nice to have figures or a section explaining what you're solving, rather than how, e.g. inputs are in the pixel space, encoder is an MLP and the TD-MPC tries to solve X ... So some part of Section E from the appendix could go to the main text.
>
> We have used Figure 1 to explain what we are solving in existing dynamics modeling methods used in RL and planning.
> We modified the starting paragraph of Section 4.2 to describe that the model-based planning and model-free RL with dynamics modeling is to provide data-efficiency in control tasks.  In the original submission, we have also mentioned in Sections 4.2.1 and 4.2.2 whether the environment observations are in pixel space or state space.
> We believe the reviewer meant Appendix C. Due to page limitations, we moved the detailed background and explanation of these methods to the Appendix. We consider these details to be supplementary rather than central to the paper's main contributions. However, in the first paragraph of Section 4.2, we encourage interested readers, especially those in the reinforcement learning (RL) community, to consult Appendix C for more details.
>
> >It is not clear to me which network architecture you used for the encoder g_{\theta}, is it another MLP?
> >As a follow-up to the above: how does the encoder quality effect your results? How do you come up with the encoder structure? Can the theory guide you here?
>
> We thank the reviewer for pointing this out. The network architecture of $g_{\theta}$ depends on the input state. Section 3.1 of the modified submission has been updated to explain this. In theory, $g_{\theta}$ needs sufficient expressivity for the prediction errors to go down as we have offloaded the task of finding eigenfunctions to it. In practice, we have observed that empirically having a standard encoder architecture like an MLP for state spaces and a CNN for pixel spaces suffices.

---

> > ### Author Response · Authors · 2023-11-21
> > **Official response to Reviewer UaZL (2/2)**
> >
> > >In the optimized Koopman matrix, are the eigenvalues sparse, or are they all nonzero/complex? Can they be made sparse and is that interesting?
> >
> > This is an interesting question. Please note that having sparser eigenvalues is the same as operating in a lower dimensional latent space as we use a diagonal Koopman structure. We provide an ablation on the dimension of latent space in Appendix I. We observe that a generally higher dimension of the latent space leads to better performance but comes with an increase in the number of parameters. This is inline with the Koopman theory. However, the performance saturates after a point. Moreover, we have observed that the learned complex eigenvalues are nonzero.
> >
> > > This is not a problem only concerning this work but in general applies to all 'Koopman-operator-inspired-approaches': [....] If not, please mention how the Koopman-operator theory guides you analytically and intuitively.
> >
> > We agree with the reviewer that Koopman theory in the context of modeling dynamics using deep neural networks usually boils down to using a linear latent dynamics model. However, there can be many formulations under different assumptions, as mentioned in Section 2. The theory can help us understand which ones we are making for the given formulation.
> >
> >  >Please mention the limitation to deterministic environments in the introduction. Is this a limitation that all Koopman-operator approaches in the literature face?
> >
> > Stochastic dynamics indeed presents a challenge to the traditional formulation of Koopman theory. However, there are extensions of classical formulations that extends it to approximately model stochastic dynamics. As mentioned in Section 7, we leave this direction for future work.
> >
> > $$$$
> >
> > [1] Schwarzer, Max, et al. "Data-Efficient Reinforcement Learning with Self-Predictive Representations." International Conference on Learning Representations. 2020.

---

> > > ### Author Response · Authors · 2023-11-22
> > > **Gentle reminder: last day of discussion phase**
> > >
> > > Dear reviewer UaZL,
> > >
> > > As today is the last day of the discussion phase, we would like to know if the reviewer has any other concerns.
> > >
> > > We appreciate the reviewer's initial positive assessment of the work and valuable comments to improve it further.
> > >
> > > Thank you,
> > > Authors

---

> ### Comment · Reviewer_UaZL · 2023-11-22
> **my score remains the same**
>
> I have read all of the points raised by the reviewers as well as the authors' replies. I would like to thank the authors for their careful rebuttal. I think this paper can be accepted, as it provides an incremental but solid improvement over the existing Koopman-operator-inspired-approaches, as far as I can tell.
>
> However I keep my score the same, because of the incremental nature of the paper and the conceptual/theoretical lack of understanding of various issues that were raised in the reviews (e.g., lack of theoretical underpinning of "Koopman-approximations", all-too-experimental explanations of several design decisions etc.)

---

### Official Review · Reviewer_xZWp · 2023-11-01

**Soundness:** 2 fair
**Presentation:** 2 fair
**Contribution:** 2 fair
**Rating:** 3
**Confidence:** 5

**Summary:**

The paper proposes an approach based on Koopman theory, which linearizes the nonlinear dynamics of the environment in a high-dimensional latent space for long term prediction. This allows for efficient parallelization of the sequential problem of long-range prediction using convolution, while considering the agent's actions at each time step. The approach also enables stability analysis and better control over gradients through time, resulting in significant improvements in efficiency and accuracy of modeling dynamics over extended horizons.

**Strengths:**

The strengths are as follows:
1) They provide extensive experimental results verifying the claims made in the better and showing better long term prediction accuracy that other recent non-Koopman NN based methods.

2) They provide numerical simulations that they Koopman based approach does not lead of instability. In other words, they show that their long term predictions do not blow up.

**Weaknesses:**

The weakness of the paper are as follows:

1) I've had the opportunity to delve into Koopman operator theory in my past research. It's noted that control-affine systems transform to a bilinear control Koopman based system in the lifted space under particular conditions, as exemplified by Theorem II.1 in the paper titled "Advantages of Bilinear Koopman Realizations for the Modeling and Control of Systems with Unknown Dynamics". It's also worth mentioning that these bilinear forms may not be universally applicable to all nonlinear systems, especially those that do not adopt the control affine form. Some recent works on Koopman operator theory have also showcased how a control-affine nonlinear system can be transitioned to a more generalized input-separable Koopman system, with bilinear and linear forms being special instances of these separable Koopman formats.

2) There are alternative NN methods, such as Neural ODE, that have consistently demonstrated promising results for long horizon predictions, in my experience. It might be valuable to consider or reference them in the context of this paper.

3) The problem of long horizon prediction for both controlled and non-controlled dynamical systems using Koopman operator has been explored before, please see the Nature publication titled "Deep learning for universal linear embeddings of nonlinear dynamics", and in subsequent studies.  I am aware that you already cited this reference but “twice” (please check)

4) While controlled systems are a primary focus in this paper, an additional section dedicated to control design could potentially improve the paper further.

5) On the topic of stability in Koopman-based learned matrices, there are several contributions, such as "Learning Stable Koopman Embeddings", and its subsequent related studies. It might be advantageous to discuss or incorporate the stability guarantees they present in the context of this paper.

6) Considering the above, a more comprehensive literature review on the Koopman operator for controlled/non-controlled dynamical systems might enhance the paper's breadth.

7) For the purpose of replication and validation, making the code available as open-source (while preserving the authors' anonymity) could be beneficial for the broader community.

**Questions:**

1) It would be interesting to see the results with NeuralODE

2) Please do a more thorough literature review on Koopman operator theory for control and modelling.

---

> ### Author Response · Authors · 2023-11-21
> **Official response to Reviewer xZWp [1/2]**
>
> We thank the reviewer for their comments. It appears to us that the overall rating is inconsistent with the review, which mentions connections to the existing Koopman theory literature and other relatively minor issues. It would be helpful if the reviewer could emphasize any (remaining) major issue to give us the chance to respond during the remaining part of the discussion period.
>
>
> > I've had the opportunity to delve into Koopman operator theory in my past research. [......] Some recent works on Koopman operator theory have also showcased how a control-affine nonlinear system can be transitioned to a more generalized input-separable Koopman system, with bilinear and linear forms being special instances of these separable Koopman formats.
>
> We thank the reviewer for pointing us to the additional relevant paper. However, our discussion of the control-affine setting and the resulting bilinear form is for the sake of completeness only. As discussed in Section 2.2, we assume a decoupling of state and control observables, leading us to equation (2). Moreover, we have cited the paper the reviewer suggested and have added more clarification on the control-affine setting assumption in Section 2.2.
>
> >There are alternative NN methods, such as Neural ODE, that have consistently demonstrated promising results for long-horizon predictions, in my experience. It might be valuable to consider or reference them in the context of this paper
>
> >It would be interesting to see the results with NeuralODE
>
> While a Neural ODE is an interesting option for sequence modelling, it is not a common baseline in RL. The reason is that the strength or advantage of a Neural ODE is in modelling “irregular’’ sequences, where time steps are not equally distanced. Existing RL benchmarks use regular sequences. Moreover, for high-dimensional data, a “latent” neural ODE is used, which is comparable to an RNN but is generally slower; for example, see [1,2]. Even in [2], where they use a Neural ODE in RL, the resulting model is “solved” to produce RNN-like updates.  In our paper, we compare to GRU, which is a stronger baseline that outperforms RNN for long-range sequence models.
>
> >The problem of long horizon prediction for both controlled and non-controlled dynamical systems using Koopman operator has been explored before;[...] I am aware that you already cited this reference but “twice” (please check)
>
> We have corrected this double-citation error in the revised article. Indeed, as the reviewer points out, we did mention this use of the Koopman operator in past work. See Sections 2.1, 2.2 and 5 for mention of relevant work to our paper. A point of contrast is that our formulation enables efficient parallel training using convolution and the diagonal Koopman matrix.
>
> >While controlled systems are a primary focus in this paper, an additional section dedicated to control design could potentially improve the paper further.
>
> We agree and would like to direct the reviewer to Section 2.2, which discusses control design in the context of Approximate Koopman. However, this section admittedly does not provide comprehensive coverage, and due to the page limit, we refer the reader to existing surveys such as [3].
>
> >On the topic of stability in Koopman-based learned matrices, there are several contributions, such as "Learning Stable Koopman Embeddings", and its subsequent related studies. It might be advantageous to discuss or incorporate the stability guarantees they present in the context of this paper.
>
> We appreciate the reviewer’s suggestion to look into [4]. We have now cited this in Section 2.1 of Preliminaries, where we discuss how Koopman theory can be used for stability analysis of dynamical systems. We want to emphasize that, as discussed in Sections 3.2.1 and 3.2.2, stability in the context of our work refers to the stability while training these dynamics models and backpropagate error from further away in time. This is different from [4], which focuses on contraction analysis for modelling stable non-linear systems and derives a Koopman formulation that respects that. Moreover, we focus on dynamical systems with control input and extend our framework to solve vanishing or exploding gradients for long-range predictions in decision-making frameworks like RL and planning.

---

> > ### Author Response · Authors · 2023-11-21
> > **Official response to Reviewer xZWp [2/2]**
> >
> > >Considering the above, a more comprehensive literature review on the Koopman operator for controlled/non-controlled dynamical systems might enhance the paper's breadth.
> >
> > >Please do a more thorough literature review on Koopman operator theory for control and modelling.
> >
> > We have referenced relevant Koopman-based literature for dynamical systems in Sections 1 and 2.1. We have also referenced related literature on Koopman operators for controlled dynamical systems in Sections 2.2 and 5. In the revised manuscript we have added a few more relevant citations based on the reviewer’s suggestions.  Due to limited space, for a more thorough discussion, we now refer the readers to comprehensive literature on Koopman theory for controlled/non-controlled dynamical systems such as [3] in Section 2.2.
> >
> > >For the purpose of replication and validation, making the code available as open-source (while preserving the authors' anonymity) could be beneficial for the broader community.
> >
> > We have added a link to an anonymous GitHub repository with the code base for the offline RL experiments with the baselines in Appendix J. This is in addition to the pseudocode of the “Diagonal Koopman Dynamics model” in Appendix J of the original submission.
> >
> > $$$$
> >
> > [1] Rubanova, Yulia, Ricky TQ Chen, and David K. Duvenaud. "Latent ordinary differential equations for irregularly-sampled time series." Advances in neural information processing systems 32 (2019).
> >
> > [2] Du, Jianzhun, Joseph Futoma, and Finale Doshi-Velez. "Model-based reinforcement learning for semi-markov decision processes with neural odes." Advances in Neural Information Processing Systems 33 (2020).
> >
> > [3] Steven L Brunton, Marko Budiši ́c, Eurika Kaiser, and J Nathan Kutz. Modern koopman theory for dynamical systems
> >
> > [4] Fletcher Fan, Bowen Yi, David Rye, Guodong Shi, and Ian R Manchester. Learning stable koopman embeddings. In 2022 American Control Conference (ACC), pp. 2742–2747. IEEE, 2022

---

> ### Author Response · Authors · 2023-11-22
> **Gentle reminder: last day of discussion phase**
>
> Dear reviewer xZWp,
>
> As we approach the conclusion of the discussion phase today, we kindly request an acknowledgement of our response. We would be happy to address any remaining major concerns the reviewer might have.
>
> Given the reviewer's initial negative assessment, it will be really valuable to know if they still have any major concerns after our rebuttal response.  This will give us the chance to respond.
>
> Thank You,
>
> Authors

---

### Author Response · Authors · 2023-11-21
**Official Comment by Authors**

We thank the reviewers for their valuable feedback. Based on the reviewer’s comments, we made the following changes to the manuscript (highlighted in blue colour):
1) Additional experiments with DSSM [1] baseline for state and reward prediction
2) Editorial changes incorporating the reviewer’s suggestions
3) Ablations for sequence length for model-based planning and model-free RL


[1] Gu, Albert, et al. "On the parameterization and initialization of diagonal state space models." Advances in Neural Information Processing Systems 35 (2022).

---

### Meta-Review · Area_Chair_Dq3n · 2023-12-07

**Metareview:**

*Summary*: This paper introduces a Koopman-based approach with a diagonal transition matrix that efficiently and stable learns nonlinear dynamics. The conventional latent space Koopman methods are modified to output the complex eigenvalues of the latent linear model, such that vanishing/exploding issues are well solved, and the computational burden is reduced over long horizons. The proposed framework is evaluated in various dynamics learning and RL tasks with competitive performance over several SOTA approaches.

*Strength*: (1) Novel ideas in the Koopman framework. The novel complex eigenvalue design significantly reduces training stability and allows better parallelization. (2) Good literature review and introduction to Koopman theory. (3) Thorough empirical studies compared to various baselines.

*Weakness*: (1) The proposed framework doesn't support stochastic dynamics settings. That is why it has not been tested for MBRL. (2) There is some gap between Koopman theory and the proposed method. The authors can further improve the paper by analyzing or showing the limitations of the proposed framework (i.e., what kind of systems it cannot learn).

**Justification For Why Not Higher Score:**

See the weakness part.

**Justification For Why Not Lower Score:**

See the strength part.

---

### Decision · Program_Chairs · 2024-01-16

Accept (poster)